# SAINT: ATTENTION-BASED POLICIES FOR DISCRETE COMBINATORIAL ACTION SPACES

## ABSTRACT

The combinatorial structure of many real-world action spaces leads to exponential growth in the number of possible actions, limiting the effectiveness of conventional reinforcement learning algorithms. Recent approaches for combinatorial action spaces impose factorized or sequential structures over sub-actions, failing to capture complex joint behavior. We introduce the Sub-Action Interaction Network using Transformers (SAINT), a novel policy architecture that represents multi-component actions as unordered sets and models their dependencies via self-attention conditioned on the global state. SAINT is permutation-invariant, sample-efficient, and compatible with standard policy optimization algorithms. In 20 distinct combinatorial environments across three task domains, including environments with nearly 17 million joint actions, SAINT consistently outperforms strong baselines. [1]

## 1 INTRODUCTION

Reinforcement learning (RL) has achieved remarkable success across a range of domains, primarily through methods designed for either small, discrete action spaces (Hessel et al., 2018; Mnih et al., 2015; van Hasselt et al., 2015; Mnih et al., 2016) or continuous control (Fujimoto et al., 2018; Haarnoja et al., 2018; Lillicrap et al., 2015; Schulman et al., 2017). Many real-world problems, however, involve action spaces that lie between these extremes. These large discrete combinatorial spaces are defined as Cartesian products of multiple subspaces, where each joint action $\mathbf{a} = (a_1, \ldots, a_A)$ consists of several coordinated sub-actions. Such settings, which arise in critical applications like traffic signal control (Rasheed et al., 2020) and drug selection (Tang et al., 2022), require learning policies that can effectively represent and reason about exponentially large, structured action spaces.

Traditional RL methods model discrete action spaces with a flat categorical policy, but this becomes intractable in combinatorial settings where the number of actions scales as $\prod_{d=1}^{A} m_d$ for $A$ sub-action dimensions with $m_d$ choices each. To mitigate this combinatorial explosion, existing approaches (Dulac-Arnold et al., 2015; Pierrot et al., 2021; Tavakoli et al., 2018; Zhang et al., 2018) rely on simplifying assumptions that constrain the representational capacity of the policy class. One family of methods (Pierrot et al., 2021; Tavakoli et al., 2018) factorizes the policy as $\pi(\mathbf{a} \mid s) = \prod_i \pi_i(a_i \mid s)$, which cannot represent interactions between sub-actions. Another class of approaches (Zhang et al., 2018) imposes a fixed autoregressive order, specifying a policy class of distributions of the form $\pi(\mathbf{a}|s) = \prod_i \pi_i(a_i|s, a_{<i})$. This introduces an arbitrary sequence over sub-actions, breaking permutation invariance and impairing learning when the imposed order misaligns with the true dependency structure. Many real-world tasks violate both the independence and fixed-order assumptions. In healthcare, for example, drug combinations can exhibit complex interaction effects — treatments may be safe individually but harmful together, motivating permutation-invariant models for combinatorial action spaces. Our work targets precisely these settings: combinatorial action spaces wherein sub-action indexing is arbitrary or only weakly meaningful, and the fundamental structure lies in sub-action interactions rather than in any prescribed ordering.

We introduce the Sub-Action Interaction Network using Transformers (SAINT), a policy architecture that learns explicit representations of combinatorial actions by treating them as unordered sets of sub-actions. Through self-attention conditioned on the global state, SAINT captures dependencies

---

[1]Code is available at `https://anonymous.4open.science/r/SAINT-6BB9`

among sub-actions to produce expressive yet tractable policies. SAINT proceeds in three stages. First, global state information is injected into initial sub-action representations. Next, self-attention (Vaswani et al., 2017) is applied over the set of state-conditioned representations to capture sub-action dependencies while preserving permutation equivariance. Finally, the representations are decoded in parallel, producing action distributions that preserve the modeled interactions while remaining computationally tractable.

We evaluate SAINT on challenging benchmark tasks, which exhibit both state-independent and state-dependent sub-action dependencies, including traffic light control (Zhang et al., 2019), navigation (Landers et al., 2024), and discretized MuJoCo locomotion tasks (Towers et al., 2024). Our results demonstrate that by learning a more expressive representation of the action space's internal structure, SAINT consistently outperforms strong factorized and autoregressive baselines, scaling to environments with nearly 17 million discrete actions. Targeted ablations validate the role of state conditioning and show that the additional cost of modeling dependencies is often offset by substantial gains in sample efficiency. Together, these findings establish that learning explicit representations of sub-action interactions is a practical and scalable approach to decision-making in complex combinatorial domains.

## 2 RELATED WORK

**Combinatorial Action Spaces** Combinatorial action spaces arise naturally in sequential decision problems such as traffic signal control, games, and resource allocation. Prior work has introduced task-specific architectures, imposed domain-specific assumptions, or exploited problem-specific structure (Bello et al., 2016; Chen et al., 2023; Delarue et al., 2020; He et al., 2015; 2016; Nazari et al., 2018; Zahavy et al., 2018; Farquhar et al., 2020). Such methods typically lack generality and require manual design effort. A parallel body of work addresses continuous control problems by discretizing the action space (Barth-Maron et al., 2018; Metz et al., 2017; Tang & Agrawal, 2020; Van de Wiele et al., 2020), which contrasts with our focus on inherently discrete action spaces with combinatorial structure.

A number of general-purpose architectures have been developed to scale RL to large combinatorial action spaces. One strategy reduces complexity by assuming conditional independence across sub-actions (Pierrot et al., 2021; Tavakoli et al., 2018), while another imposes an autoregressive order (Zhang et al., 2018). These approaches improve tractability but either ignore dependencies among sub-actions or introduce arbitrary orderings that break permutation symmetry. Retrieval-based methods such as Wolpertinger (Wol-DDPG) (Dulac-Arnold et al., 2015) scale to large discrete spaces by embedding and pruning candidate actions, but similarly fail to capture the joint structure of unordered sub-actions (Chen et al., 2023). These limitations motivate architectures that can represent dependencies across sub-actions while preserving permutation invariance.

**Transformers for Action Representation** Efforts to use Transformers for action space modeling have largely focused on sequential representations. RT-1 (Brohan et al., 2022) and RT-2 (Zitkovich et al., 2023) tokenize robot control trajectories and decode action tokens autoregressively, while Q-Transformer (Chebotar et al., 2023) autoregresses across action dimensions within a timestep. Learned tokenizers such as FAST (Pertsch et al., 2025) compress high-frequency control signals into vocabularies for vision–language–action (VLA) training. Extensions of trajectory-based models (Chen et al., 2021; Shang et al., 2022), interleave state and action tokens or design state-aware tokenizations to handle multi-component actions. Together, these methods establish the efficacy of Transformers for action encoding, but have been mostly designed for continuous control tasks in offline RL or VLA settings that are not directly comparable to the combinatorial domains we study. Moreover, they largely rely on sequential decompositions that impose arbitrary order and obscure permutation symmetry.

## 3 PRELIMINARIES

**Combinatorial Action Spaces** We consider RL problems formalized as a Markov Decision Process, defined by the tuple $\mathcal{M} = \langle \mathcal{S}, \mathcal{A}, p, r, \gamma, \mu \rangle$. Here, $\mathcal{S}$ denotes the state space, $\mathcal{A}$ the action space, $p(s' \mid s, a)$ the transition function, $r(s, a)$ the reward function, $\gamma \in [0, 1]$ the discount factor, and

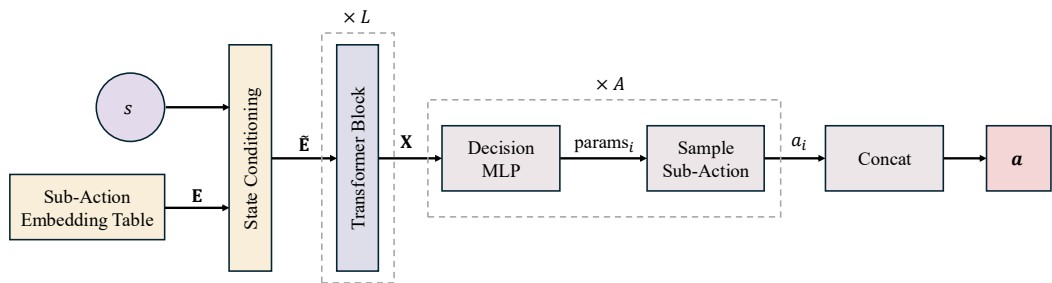

Figure 1: Overview of SAINT. Initial sub-action embeddings are conditioned on the global state **s** to produce state-aware representations. Stacked Transformer blocks then model dependencies among sub-actions. The resulting context-aware representations are passed to independent Decision MLPs, which output per-sub-action policy distributions used for factorized sampling.

$\mu$ the initial state distribution. A policy $\pi$ maps states to distributions over actions, $\pi : \mathcal{S} \to \mathbb{P}(\mathcal{A})$, which defines the agent's behavior in the environment.

In this work we assume actions have an explicit compositional structure. Specifically, the action space is a product of sub-action domains, $\mathcal{A} = \mathcal{A}_1 \times \cdots \times \mathcal{A}_A$, where each sub-action space $\mathcal{A}_i$ is a discrete set. A single action thus comprises $A$ sub-decisions, $\mathbf{a} = (a_1, \ldots, a_A)$, with each $a_i \in \mathcal{A}_i$. This representation gives rise to high-dimensional action spaces with potentially complex dependencies among sub-actions.

**Attention**   Attention is a general computational primitive that allows a model to selectively aggregate information from a set of inputs based on learned relevance scores (Vaswani et al., 2017). Formally, given a set of queries $Q \in \mathbb{R}^{n_q \times d}$, keys $K \in \mathbb{R}^{n_k \times d}$, and values $V \in \mathbb{R}^{n_k \times d}$, the scaled dot-product attention computes output $\mathrm{Attn}(Q, K, V) = \mathrm{softmax}(QK^\top / \sqrt{d})V$. Multi-head self-attention extends this by learning multiple independent projections and aggregating their outputs, enabling the model to capture different types of interactions in parallel. Transformers, which stack layers of multi-head self-attention with feedforward components, have been widely adopted in domains requiring flexible modeling of structured dependencies.

# 4   SUB-ACTION INTERACTION NETWORK USING TRANSFORMERS (SAINT)

We introduce the Sub-Action Interaction Network using Transformers (SAINT), a policy architecture that learns a state-conditioned, permutation-equivariant representation of the action space, enabling efficient computation of expressive action distributions. The SAINT architecture comprises three stages: (1) state conditioning, which injects global state information into sub-action representations; (2) interaction modeling, which applies self-attention to model higher-order relationships among sub-actions while preserving permutation equivariance; and (3) action decoding, which transforms each sub-action representation into a distribution over its discrete choices, with all sub-actions decoded in parallel to maintain tractability. An overview of the SAINT architecture is shown in Figure 1.

## 4.1   STATE CONDITIONING

SAINT represents each sub-action $i \in \{1, \ldots, A\}$ with a learnable embedding vector $\mathbf{e}_i = \mathrm{Embed}(i) \in \mathbb{R}^d$, drawn from a table $\mathrm{Embed} \in \mathbb{R}^{A \times d}$. With $d$ treated as a fixed hyperparameter shared across all sub-actions, each sub-action is represented by a $d$-dimensional embedding independent of its original cardinality $|\mathcal{A}_i|$, yielding a shared space for uniform processing by the subsequent Transformer layers.

Because sub-action identity alone is insufficient for decision-making, SAINT augments each base embedding $\mathbf{e}_i$ with information from the global state $\mathbf{s} \in \mathbb{R}^{d_s}$, enabling dependencies to be modeled in a state-aware manner. While several conditioning mechanisms such as cross-attention or concatenation are possible, we adopt Feature-wise Linear Modulation (FiLM) (Perez et al., 2018), which we found to be effective and parameter-efficient (see Appendix C). Notably, FiLM preserves the

---

**Algorithm 1** Policy Learning with the SAINT Architecture

---

1: Initialize SAINT policy $\pi_\theta$ and value network $V_\phi$
2: **for** each training iteration **do**
3:      Collect a batch of transitions $(\mathbf{s}_t, \mathbf{a}_t, r_t, \text{done}_t)$ by executing $\pi_\theta$
4:      Compute return $R_t$ and weighting term $w_\Phi(\mathbf{s}_t, \mathbf{a}_t, R_t)$ for each transition
5:      Compute policy log-probabilities $\ell_t \leftarrow \text{LOGPROBS}(\mathbf{s}_t, \mathbf{a}_t)$          $\triangleright$ See function below
6:      Update policy $\theta$ by ascending the objective $\mathbb{E}_t[w_\Phi(\mathbf{s}_t, \mathbf{a}_t) \cdot \ell_t]$
7:      Update value function $\phi$ by descending the loss $\mathbb{E}_t[(V_\phi(\mathbf{s}_t) - R_t)^2]$
8: **end for**

9: **function** LOGPROBS($\mathbf{S}, \mathbf{A}_{\text{taken}}$)
10:      Get sub-action embeddings $\mathbf{E} \leftarrow [\text{Embed}(1), \ldots, \text{Embed}(A)]^\top$
11:      Inject state information $\tilde{\mathbf{E}} \leftarrow \text{StateCondition}(\mathbf{S}, \mathbf{E})$
12:      Model interactions $\mathbf{X} \leftarrow \text{TransformerBlocks}(\tilde{\mathbf{E}})$
13:      Get logits for each sub-action $\text{Logits}_i \leftarrow f_i(\mathbf{X}_{[:,i]})$ for $i = 1, \ldots, A$
14:      Compute log-probabilities $\log \mathbf{P}_{[:,i]} \leftarrow \log \pi_i(\mathbf{A}_{\text{taken}[:,i]} \mid \text{Logits}_i)$ for $i = 1, \ldots, A$
15:      **return** $\sum_{i=1}^A \log \mathbf{P}_{[:,i]}$
16: **end function**

---

$d$-dimensional width of each sub-action embedding, introducing no additional projection dimensions. Its prior success in incorporating state information in RL (Brohan et al., 2022) further supports this choice.

An MLP $g : \mathbb{R}^{d_s} \rightarrow \mathbb{R}^{2d}$ processes the global state $\mathbf{s}$ once to produce FiLM parameters $(\boldsymbol{\gamma}, \boldsymbol{\beta}) = g(\mathbf{s})$, which are then applied uniformly to all sub-action embeddings via an affine transformation:

$$\tilde{\mathbf{e}}_i = \boldsymbol{\gamma} \odot \mathbf{e}_i + \boldsymbol{\beta} .$$

### 4.2 INTERACTION MODELING

The matrix of state-aware sub-action representations $\tilde{\mathbf{E}} \in \mathbb{R}^{A \times d}$ is then processed by a stack of $L$ Transformer blocks, with positional encodings omitted to preserve permutation equivariance. Letting $\mathbf{X}^{(0)} = \tilde{\mathbf{E}}$, each block $\ell = 1, \ldots, L$ performs multi-head self-attention followed by a feed-forward network (FFN). Specifically, queries, keys, and values are obtained by linear projections of the previous layer's output:

$$\mathbf{Q}, \mathbf{K}, \mathbf{V} = \mathbf{X}^{(\ell-1)} W^Q, \ \mathbf{X}^{(\ell-1)} W^K, \ \mathbf{X}^{(\ell-1)} W^V ,$$

which are then used in scaled dot-product attention to model interactions among sub-actions. The attention output is then passed through a position-wise FFN applied independently to each sub-action embedding. This design allows SAINT to model state-conditioned dependencies between sub-actions while maintaining permutation equivariance.

### 4.3 ACTION DECODING

In the final stage, each context-aware sub-action representation $\mathbf{x}_i$ is passed through a sub-action-specific decision MLP, $f_i : \mathbb{R}^d \rightarrow \mathbb{R}^{K_i}$, which outputs a vector of $K_i$ logits. These logits are then transformed into a probability distribution over the $K_i$ discrete choices for sub-action $i$ via the softmax function. The resulting policy for sub-action $i$ is thus given by:

$$\pi_i(a_i \mid \mathbf{s}) = \text{Categorical}(\text{softmax}(f_i(\mathbf{x}_i))) .$$

Because each sub-action representation $\mathbf{x}_i$ from the Interaction Modeling stage is conditioned on the global state and incorporates information from the other sub-actions, the policy can be expressed as independent sub-action distributions without loss of modeling capacity, preserving tractability in combinatorial spaces where representing the full joint distribution would be infeasible.

### 4.4 COMPATIBILITY WITH RL ALGORITHMS

The SAINT architecture is compatible with any RL algorithm for which the actor objective maximizes the log-likelihood of sampled joint actions $\mathbf{a}$, weighted by some functional $w_\Phi(s, \mathbf{a})$:

$$\max_\theta \ \mathbb{E}_{(s,\mathbf{a})\sim\mu}\big[w_\Phi(s, \mathbf{a}) \log \pi_\theta(\mathbf{a} \mid s)\big],$$

where $\mu$ denotes the sampling distribution over $(s, \mathbf{a})$, arising either from an online policy or from a fixed dataset in the offline setting. The weight $w_\Phi(s, \mathbf{a}) \geq 0$ is an algorithm-dependent scalar, such as an advantage term or a score derived from $Q_\Phi$.

Compatible methods include standard online algorithms such as PPO (Schulman et al., 2017) and A2C (Mnih et al., 2016), as well as offline approaches such as IQL (Kostrikov et al., 2021) and AWAC (Nair et al., 2020). SAINT also supports selection-based actor updates as in BCQ (Fujimoto et al., 2019), where the policy is trained on candidate joint actions drawn from a dataset or proposal distribution. SAINT remains compatible even when $w_\Phi(s, \mathbf{a})$ is computed with a factorized critic, since the critic is used only to produce a scalar weight for each sampled joint action from $\mu$. The actor is always updated toward the observed joint action $\mathbf{a}$, not an action reconstructed or optimized over by the critic.

Incompatibility arises when the actor objective requires global operations over the entire combinatorial action space, such as $\mathbb{E}_{\mathbf{a}'\sim\pi*\theta}[Q_\Phi(s, \mathbf{a}')]$ or $\max_{\mathbf{a}'} Q_\Phi(s, \mathbf{a}')$. These operations are computationally intractable unless $Q_\Phi$ is factorized; however, this changes the structure of the actor target, decomposing it into uncoordinated per-dimension terms and discarding cross-dimensional structure. This breaks alignment with SAINT's objective of modeling dependencies among sub-actions. SAINT's learning procedure is provided in Algorithm 1.

## 5 EXPERIMENTAL EVALUATION

Our experiments evaluate the efficacy of different action representations for modeling complex sub-action interactions across three regimes: (1) primarily state-independent interactions, (2) state-dependent interactions, and (3) weak interactions with complex dynamics. Results are presented in Sections 5.1, 5.2, and 5.3, respectively. Section 5.4 evaluates SAINT in the offline RL setting. Section 5.5 analyzes key architectural choices, quantifying the trade-off between representational power and computational cost, and robustness to Transformer structural parameters.

We compare SAINT to four baselines reflecting the standard representational assumptions for combinatorial action spaces: (1) a factorized policy (Tavakoli et al., 2018), assuming fully independent sub-actions; (2) an autoregressive model (Zhang et al., 2018), imposing a fixed sequential order; (3) Wol-DDPG (Dulac-Arnold et al., 2015), using a continuous embedding; and (4) a flat RL algorithm, which learns a monolithic representation of the full action space without exploiting its combinatorial structure. Results are averaged over five random seeds.

These four baselines instantiate the dominant structural assumptions used to scale RL to large combinatorial action spaces. The flat policy corresponds to a monolithic model that ignores compositional structure and treats each joint action as an atomic symbol. The factorized policy enforces independent per-dimension decisions, preventing it from modeling necessary coordination between sub-actions. The autoregressive model imposes a fixed sequential ordering over sub-actions, which can be misaligned with the true, permutation-invariant dependency structure. Wol-DDPG embeds each joint action as a single continuous vector, collapsing the internal structure needed to capture interactions among sub-actions. Our experiments in Sections 5.1-5.4 are designed to test whether these structural assumptions remain sufficient when sub-action indexing is arbitrary or only weakly meaningful, or whether a set-based alternative such as SAINT is required.

### 5.1 STATE-INDEPENDENT SUB-ACTION DEPENDENCIES

To evaluate SAINT in environments where sub-action dependencies are primarily state-independent, we use the CityFlow traffic control benchmark (Zhang et al., 2019), where each action corresponds to simultaneous phase decisions across multiple intersections. While coordination is necessary to achieve global traffic efficiency, the structure of these dependencies remains largely unchanged across states.

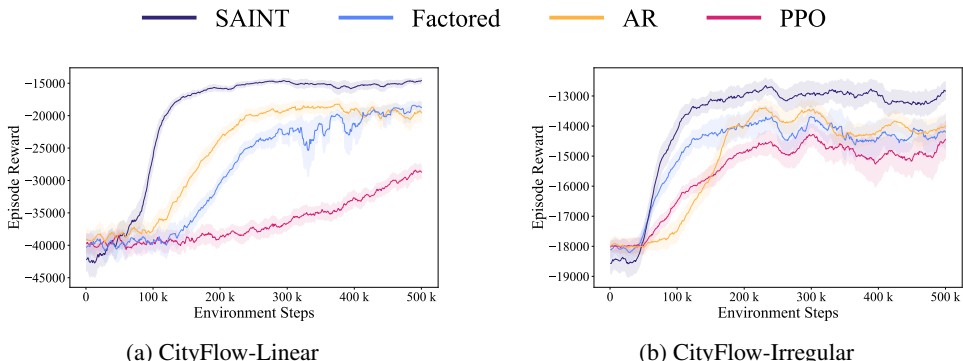

(a) CityFlow-Linear

(b) CityFlow-Irregular

Figure 2: Learning curves show that SAINT outperforms all baselines in both learning speed and final reward, across both the CityFlow-Linear and CityFlow-Irregular environments.

We consider two configurations: (1) CityFlow-Linear, where three traffic signals are arranged in a row, yielding 729 possible joint signal combinations; and (2) CityFlow-Irregular, with four closely spaced, asymmetric intersections and varying road capacities, resulting in 375 valid joint action combinations. These networks are illustrated in Appendix A.1.

We adopt PPO as the standard RL algorithm in this setting. SAINT, the pure factorized baseline, and the AR baseline, use the same PPO implementation and hyperparameters, isolating the impact of architectural differences. Wol-DDPG performs poorly in this environment and is excluded from the main learning curves; full training curves, including Wol-DDPG, are included in Appendix A.2. Wol-DDPG's ineffectiveness aligns with prior observations that it is ill-suited to environments with unordered sub-actions (Chen et al., 2023).

The results in Figure 2, demonstrate the advantage of SAINT's set-based action representation. In both environments, SAINT learns faster and achieves higher final performance. The factorized approach is structurally incapable of modeling coordination between intersections and thus performs poorly. The autoregressive model is restricted by its fixed sequential prior, which is misaligned with the unordered nature of traffic-signal control. The flat PPO baseline, forced to learn a monolithic representation without exploiting combinatorial structure, fails to learn an effective policy.

## 5.2 STATE-DEPENDENT SUB-ACTION DEPENDENCIES

Next, we evaluate SAINT in environments where sub-action dependencies are strongly state-dependent. In these domains, the relationships among sub-actions vary significantly with the environment state, requiring policies to model context-sensitive joint decisions. To study this setting, we use the Combinatorial Navigation Environment (CoNE) (Landers et al., 2024), a configurable high-dimensional control domain designed to evaluate policy architectures under large, discrete, and structured action spaces.

The agent begins at a fixed origin $s_0$ and must reach a predefined goal $g$. At each timestep, it selects a joint action by activating multiple discrete sub-actions, each corresponding to movement along a distinct dimension of the environment. These sub-actions are executed in parallel to produce a single composite transition. The agent receives a reward $r = -\rho(s, g)$ at each step based on the Euclidean distance to the goal. Episodes terminate either upon reaching the goal (reward $+10$) or entering a terminal failure state (pit), which incurs a penalty of $r = -10 \cdot \rho(s_0, g)$ to discourage reward hacking through early failure.

In CoNE, both the action and state spaces grow exponentially with dimensionality: the number of joint actions scales as $|\mathcal{A}| = 2^{2D}$, and the number of states as $|\mathcal{S}| = M^D$ in a $D$-dimensional environment with $M$ positions per axis. In our largest setting, the environment comprises over 200 million states and nearly 17 million joint actions per state. Beyond scale, CoNE introduces strong sub-action dependencies — some combinations enable efficient movement, others cancel out, and some lead to catastrophic failure. Crucially, these sub-action interactions are highly state-dependent; a combination that is optimal in one state may lead to a pit in another, making effective decision-making

| $|\mathcal{A}|$ | SAINT | Factored | AR | Wol-DDPG | A2C |
|---|---|---|---|---|---|
| $\sim$16k | -8.3 $\pm$ 0.0 | $-11.9 \pm 1.0$ | -8.3 $\pm$ 0.0 | $-586.2 \pm 62.4$ | $-593.7 \pm 51.7$ |
| $\sim$65k | -9.9 $\pm$ 0.6 | $-45.8 \pm 16.7$ | $-22.3 \pm 8.1$ | $-691.5 \pm 51.3$ | $-641.0 \pm 78.0$ |
| $\sim$260k | -12.5 $\pm$ 1.6 | $-51.6 \pm 23.0$ | $-20.4 \pm 2.6$ | $-712.0 \pm 64.0$ | $-756.3 \pm 37.2$ |
| $\sim$1M | -12.2 $\pm$ 1.3 | $-50.9 \pm 20.7$ | $-28.6 \pm 3.1$ | $-674.5 \pm 25.3$ | $-801.1 \pm 14.4$ |
| $\sim$4M | -14.4 $\pm$ 0.8 | $-36.6 \pm 6.0$ | $-28.0 \pm 2.7$ | $-929.3 \pm 43.3$ | $-846.6 \pm 4.2$ |
| $\sim$17M | -13.4 $\pm$ 2.6 | $-44.1 \pm 16.2$ | $-33.9 \pm 10.8$ | $-873.2 \pm 59.7$ | – |

Table 1: Performance in CoNE as action dimensionality increases. SAINT consistently achieves the highest reward across all action space sizes. Factorized and autoregressive baselines plateau at substantially lower reward levels, while Wol-DDPG and A2C fail to learn viable policies.

highly sensitive to global context. CoNE is highly configurable, allowing us to systematically vary the number of dimensions and pit density to assess the impact of increasing action space size and sub-action dependence.

We adopt A2C as the standard RL algorithm in this setting. SAINT, the pure factorized baseline, and the AR baseline use identical A2C implementations to ensure a controlled comparison.

**Varying Dimensionality**    To evaluate SAINT's effectiveness as the number of possible actions increases, we scale the dimensionality of CoNE from 7 (yielding over 16 thousand possible action combinations) to 12 (with nearly 17 million combinations). In CoNE environments without pits, the agent can learn a trivial policy, as the optimal solution involves selecting the same action in every state. Thus, to introduce meaningful complexity and prevent this degenerate behavior, we place pits in 25% of interior states.

The results in Table 1 show that SAINT maintains strong performance as action dimensionality increases, significantly outperforming all baselines at every scale. While the factorized and autoregressive baselines achieve modest performance in lower dimensions, their performance degrades or plateaus as the number of sub-actions grows. This suggests their fixed representational priors — assuming either complete independence or a single fixed order — are insufficient to capture the complex interactions that emerge at scale. SAINT's relative advantage increases in the largest settings, where it maintains low variance and stable performance. Wol-DDPG and A2C perform poorly throughout, highlighting their inability to form a tractable and meaningful representation of large, unordered action spaces. Note that A2C is omitted at the highest dimensionality due to the computational intractability of modeling the full joint action space with a flat categorical policy. Full learning curves are provided in Appendix B.2.

**Varying Dependence**    To assess SAINT's robustness to different levels of sub-action dependence, we incrementally increased the number of pits in the 12-dimensional CoNE environment. Higher pit densities impose stronger coordination requirements, as more sub-action combinations must be carefully selected to avoid pits. To ensure that a valid path from the start state to the goal always exists, pits were placed only in interior (non-boundary) states. We generate environments with 10%, 25%, 50%, 75%, and 100% of interior states occupied by pits. Note that even in the 100% setting, all boundary states remain pit-free, guaranteeing the existence of at least one (possibly inefficient) path to the goal region. We exclude A2C from this experiment due to the computational intractability of modeling such a large discrete action space (nearly 17 million actions) with a flat categorical distribution.

The results in Table 2 demonstrate that SAINT is more robust to increasing sub-action dependence than all baselines. As pit density increases from 10% to 100%, SAINT maintains high performance with low variance, while the factorized and autoregressive baselines generally degrade. Wol-DDPG was unable to learn meaningful policies at any pit density. These results highlight SAINT's ability to capture complex, context-sensitive dependencies between sub-actions that are critical in many real-world combinatorial environments. Full learning curves for these results are provided in Appendix B.3.

| Pit % | SAINT | Factored | AR | Wol-DDPG |
|---|---|---|---|---|
| 10 | -13.5 ± 2.7 | −44.1 ± 16.2 | −33.9 ± 10.8 | −873.2 ± 59.7 |
| 25 | -15.5 ± 2.6 | −33.3 ± 4.1 | −42.2 ± 10.7 | −822.3 ± 36.8 |
| 50 | -18.4 ± 1.2 | −78.6 ± 29.7 | −38.8 ± 4.4 | −863.0 ± 22.9 |
| 75 | -19.7 ± 0.0 | −58.0 ± 11.9 | −27.8 ± 3.1 | −841.1 ± 39.5 |
| 100 | -19.7 ± 0.0 | −54.8 ± 7.1 | −28.6 ± 4.1 | −879.5 ± 33.4 |

Table 2: Performance in 12-D CoNE ($\sim$ 17M actions) as sub-action dependence increases (controlled via pit density). SAINT consistently achieves the highest rewards across all settings and remains robust even as the sub-actions become highly dependent. Other methods degrade more rapidly, especially the factorized baseline. Wol-DDPG failed to learn meaningful policies in this setting.

## 5.3 WEAK SUB-ACTION DEPENDENCIES WITH COMPLEX DYNAMICS

To evaluate SAINT in environments where sub-action dependencies are relatively weak but the underlying dynamics are complex, we consider discretized variants of the HalfCheetah, Hopper, and Walker2D MuJoCo locomotion tasks (Towers et al., 2024). In these environments, each continuous joint control signal is discretized into 11 bins, yielding large, structured action spaces while retaining the rich temporal and physical dynamics of the original tasks. Although the discretized action spaces are combinatorially large, prior work (Beeson et al., 2024) suggests that the dependencies among sub-actions are relatively weak in these domains. This setting thus provides a useful test of SAINT's generality, elucidating whether the architectural overhead of self-attention remains beneficial when sub-action dependencies are weak or whether simpler factorized policies suffice.

Given the limitations of PPO and Wol-DDPG identified in Sections 5.1 and 5.2, we restrict our comparison to the pure factorized and autoregressive baselines, using identical PPO implementations to ensure a controlled evaluation.

As shown in Figure 3, SAINT matches baseline performance in HalfCheetah and achieves faster learning and higher returns in Hopper and Walker2D. This demonstrates that even when sub-action dependencies are weak, learning a set-based representation of the action space gives an advantage over the rigid assumptions of factorization or a fixed autoregressive order.

## 5.4 OFFLINE RL

Finally, we evaluate whether SAINT can be used effectively as a policy architecture in offline RL. Specifically, we use the `medium-expert` datasets from the discretized DM Control tasks cheetah run, finger spin, humanoid stand, quadruped walk, and dog trot introduced by Beeson et al. (2024). Across these environments, the number of sub-actions ranges from six to 39.

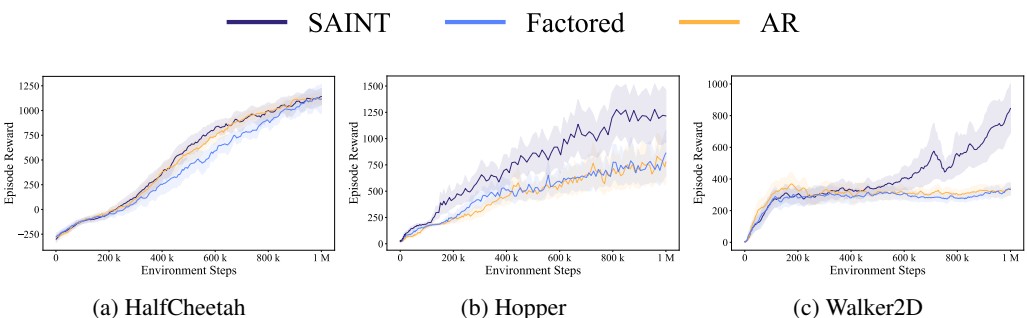

(a) HalfCheetah      (b) Hopper      (c) Walker2D

Figure 3: Performance in discretized MuJoCo environments. While results are similar across methods in HalfCheetah, SAINT outperforms factorized and autoregressive baselines in Hopper and Walker2D, demonstrating its ability to handle complex action spaces even when sub-action dependencies are relatively weak.

| Task | SAINT | Factored | AR |
|------|-------|----------|-----|
| cheetah | 676.1 $\pm$ 30.9 | 629.6 $\pm$ 36.5 | 629.6 $\pm$ 38.8 |
| finger | 809.6 $\pm$ 29.0 | 692.3 $\pm$ 71.7 | 762.1 $\pm$ 50.9 |
| humanoid | 676.5 $\pm$ 48.2 | 594.1 $\pm$ 47.7 | 592.4 $\pm$ 58.5 |
| quadruped | 851.5 $\pm$ 32.5 | 835.2 $\pm$ 52.0 | 692.2 $\pm$ 99.7 |
| dog | 586.1 $\pm$ 30.5 | 415.2 $\pm$ 40.4 | 423.3 $\pm$ 72.6 |
| **Average** | 720.0 | 633.3 | 619.9 |

Table 3: Mean $\pm$ std performance on offline DM Control tasks with BCQ variants.

In this section, we report results using Batch-Constrained Q-learning (BCQ) (Fujimoto et al., 2019) as the offline RL objective. Additional offline evaluations with AWAC and IQL are provided in Appendix C. As in Section 5.3, we limit our comparison to the pure factorized and autoregressive baselines.

As shown in Table 3, SAINT-BCQ achieves the strongest performance across all domains, demonstrating that explicitly modeling sub-action interactions improves policy quality even when learning is restricted to logged trajectories.

## 5.5 ANALYSES AND ABLATIONS

To evaluate SAINT's design choices and robustness, we conduct three analyses. First, we compare FiLM-based state conditioning to alternative mechanisms. Second, we assess the trade-off between representational power and computational cost, quantifying how modeling sub-action interactions affects sample efficiency. These evaluations are performed in the CityFlow-Irregular environment and in the 10-dimensional CoNE setting with pits occupying 25% of interior states. Finally, we test SAINT's robustness to architectural hyperparameters using the CityFlow-Irregular environment.

**State Conditioning** We compare SAINT's FiLM-based state conditioning to four alternatives: (1) applying cross-attention to the state before self-attention, (2) applying cross-attention to the state after self-attention, (3) interleaving cross-attention and self-attention layers, and (4) appending the state as an additional token within the sub-action self-attention block. As shown in Appendix C, FiLM achieves higher final performance and more stable training in CityFlow, and performs at least as well as the alternatives in CoNE. These results indicate that while multiple conditioning mechanisms are effective, FiLM provides a consistent performance advantage and stabilizes training.

**Representational Power vs. Sample Efficiency** To isolate the computational overhead of modeling sub-action dependencies via self-attention, we compare SAINT's runtime to that of the pure factorization baseline. We also evaluate a variant of SAINT, called SAINT-IP, that replaces standard self-attention with an inducing point mechanism (Lee et al., 2019), which approximates full attention using a fixed set of learned summary vectors. This technique reduces the quadratic cost of attention by attending first from the inducing points to the inputs, and then from the inputs back to the summaries. All experiments were conducted on a single NVIDIA A40 GPU using Python 3.9 and PyTorch 2.6. We report wall-clock time per training episode in seconds, averaged over 5 runs.

As shown in Table 4, SAINT requires more training time than the pure factorization baseline, reflecting the added cost of modeling sub-action dependencies with self-attention. SAINT-IP incurs further overhead from the Induced Set Attention Block (ISAB), which performs two attention passes per layer, compared to one in standard self-attention. While the number of sub-actions in our environments is nontrivial, it remains small relative to domains such as large language modeling or 3-D vision, for which inducing points were introduced (Lee et al., 2019). Consequently, the quadratic cost of full self-attention is not prohibitive, and ISAB's asymptotic advantage does not yield runtime benefits in practice.

However, in practical settings efficiency is better measured by wall-clock time to reach a target return. The "time to factored performance" metric shows that SAINT and SAINT-IP reach the factorized baseline's asymptotic return in less than one-third of the time in CityFlow and about 30% faster in CoNE. This indicates that the added computation of explicit action representations is outweighed by

|  | SAINT | SAINT-IP | Factored |
|---|---|---|---|
| **CityFlow** | | | |
| Total training time | 5204.2 | 5412.4 | 3936.2 |
| Time to factored performance | 1088.53 | 1087.49 | 3936.2 |
| **CoNE** | | | |
| Total training time | 966.5 | 980.9 | 535.9 |
| Time to factored performance | 395.76 | 504.97 | 535.9 |

Table 4: Total wall-clock training time and time to reach the factored baseline's final performance, both measured in seconds. "Total training time" reflects the full duration of training, while "time to factored performance" measures how quickly each method reaches that baseline.

faster learning — by modeling the true dependency structure, SAINT achieves stronger policies in less time.

**Robustness to Architectural Hyperparameters** We evaluated SAINT's sensitivity to architectural hyperparameters by sweeping the number of self-attention blocks $\{1, 3, 5\}$ and attention heads $\{1, 2, 4, 8\}$, for a total of twelve configurations. As shown in Appendix F, performance varied within a narrow range — the best setting (3 blocks $\times$ 1 head) outperformed the weakest (1 block $\times$ 8 heads) by only $\sim 7\%$. Even the weakest configuration exceeded all baselines, underscoring SAINT's robustness to attention depth and head count.

## 6 DISCUSSION AND CONCLUSION

We introduce SAINT, a policy architecture that treats learning in large discrete action spaces as a representation learning problem. Instead of assuming conditional independence or a fixed ordering, SAINT learns a state-conditioned, permutation-equivariant set representation of the combinatorial action space. Self-attention models the interactions within this set, yielding expressive and tractable policies for combinatorial domains.

While SAINT achieves strong performance, several limitations remain. Self-attention introduces a higher per-step computational cost than purely factorized baselines. Section 5.5 shows that this cost is often offset by improved sample efficiency, but lighter-weight attention variants such as sparse attention (Child et al., 2019) could benefit resource-constrained settings. Our analyses also validate FiLM as an effective state-conditioning mechanism, yet performance in new domains may depend on the capacity of this network, motivating exploration of more expressive state-injection methods such as those proposed in multi-agent RL (Iqbal & Sha, 2019). Next, SAINT is designed for domains wherein a joint action is naturally represented as a set of parallel sub-actions for which indexing is arbitrary or weakly structured. In settings with highly structured and known priors, a fully permutation-equivariant prior may not be the most effective representation. In such settings hybrid architectures that combine structured embeddings with partial equivariance are a promising direction for future work. Finally, although our experiments assume a fixed set of sub-actions, many real-world domains, such as road closures or reconfigurable network topologies, involve dynamically changing action sets. Because SAINT represents actions as an unordered set, it naturally supports such variability via masking. Systematically evaluating this capability is an important research direction.

In this work we show that learning explicit representations of sub-action interactions is an effective and practical approach for control in combinatorial action spaces. SAINT represents actions as unordered sets and applies self-attention to capture sub-action dependencies, yielding expressive yet tractable policies. This modeling delivers substantial gains in sample efficiency, accelerating convergence to high-performing policies. In environments with up to 17 million joint actions, SAINT consistently outperforms baselines that assume independence, impose ordering, or learn flat policies, demonstrating the effectiveness of modeling sub-action interactions for scalable combinatorial control.

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

# A  LEARNING IN CITYFLOW

## A.1  ENVIRONMENTAL SETUP

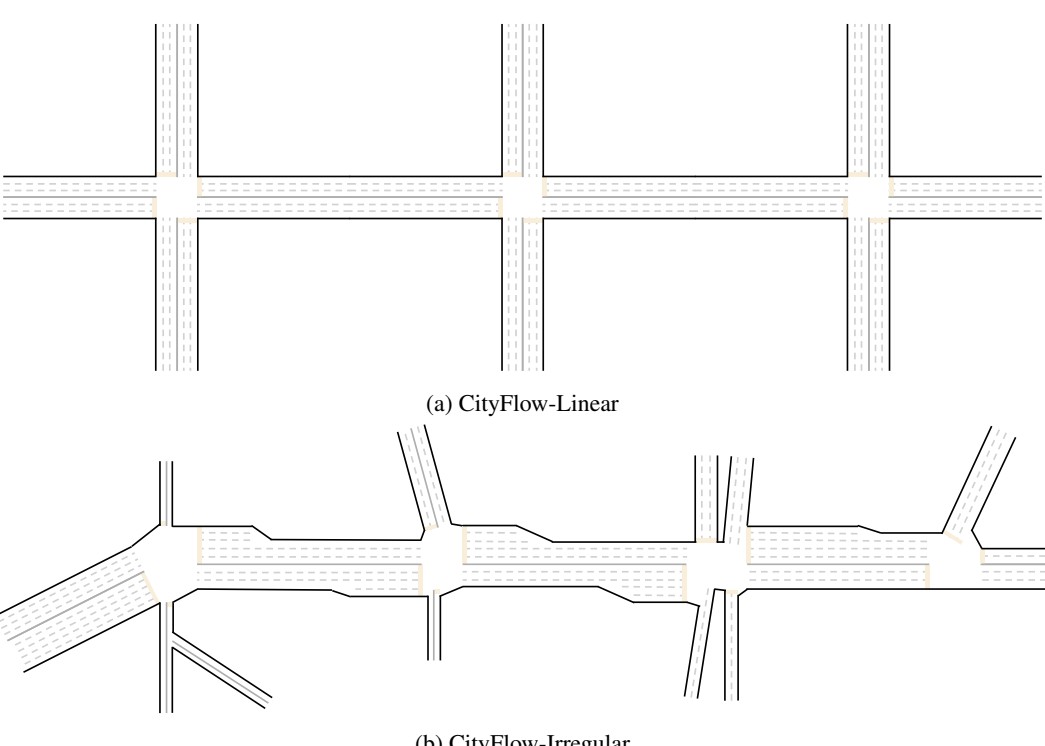

(a) CityFlow-Linear

(b) CityFlow-Irregular

Figure 4: Visualizations of the two CityFlow traffic control configurations used in our experiments. CityFlow-Linear (Figure 4a) has three intersections arranged in a row, yielding 729 possible joint actions. CityFlow-Irregular (Figure 4b) has 375 joint actions but exhibits greater coordination demands and more diverse traffic interactions.

In both CityFlow-Linear (Figure 4a) and CityFlow-Irregular (Figure 4b), the state is represented as a flat integer vector, in which each value indicates the number of waiting vehicles on an incoming lane and its paired outgoing lane. The reward at each step is the negative of the average "pressure" across intersections, where an intersection's pressure is defined as the absolute difference between its total incoming and outgoing vehicle counts. Pressure is a standard metric in traffic signal control literature, commonly used to quantify imbalance in intersection flow Lioris et al. (2016); Varaiya (2013).

## A.2 LEARNING CURVES INCLUDING WOL-DDPG

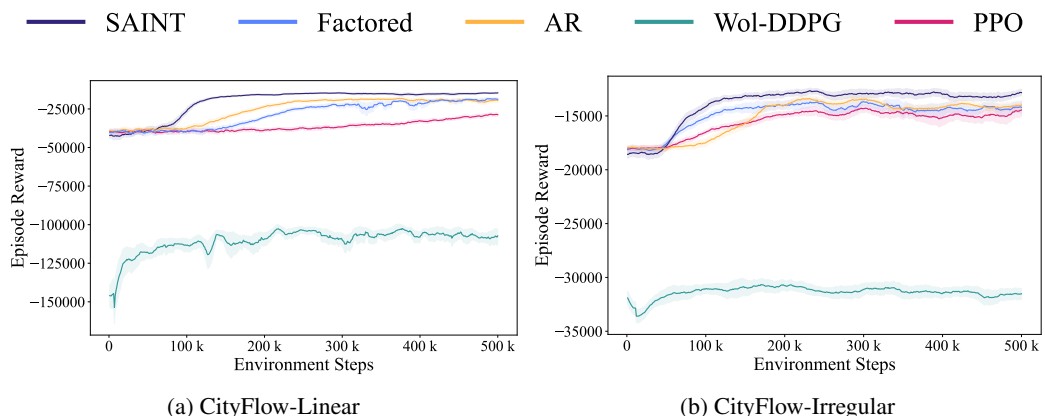

(a) CityFlow-Linear

(b) CityFlow-Irregular

Figure 5: Full learning curves for all baselines in CityFlow, including Wol-DDPG. Wol-DDPG performs poorly, consistent with its known limitations in environments with unordered sub-actions.

Figure 5 presents the full training curves for all methods in the CityFlow environments, including Wol-DDPG. As noted in Section 5.1, Wol-DDPG consistently underperforms relative to other methods. This poor performance is consistent with prior findings Chen et al. (2023), which show that Wol-DDPG is ill-suited to settings with unordered sub-actions. Learning curves excluding Wol-DDPG are presented in Figure 2.

# B   CoNE Learning Curves

## B.1   Environmental Setup

The Combinatorial Navigation Environment (CoNE) Landers et al. (2024) is designed to evaluate RL algorithms in settings with high-dimensional, combinatorial action spaces and strong, state-dependent sub-action dependencies. In CoNE, actions are formed by simultaneously selecting discrete sub-actions, each specifying movement along a different dimension. These sub-actions are executed in parallel to produce a composite transition, which may advance the agent toward the goal or result in failure by entering a pit.

CoNE supports scaling along two axes, action dimensionality and pit density. As the number of dimensions increases, both the state and action spaces grow exponentially; our largest configuration contains over 200 million states and nearly 17 million discrete joint actions per state. In CoNE, sub-action interactions are complex: some combinations are efficient, others cancel each other out, and many must be avoided. These dependencies are highly state-sensitive, requiring effective decision-making to account for both structure and context.

To our knowledge, no other existing benchmarks offer the combination of large-scale action spaces and tunable sub-action dependencies found in CoNE. Popular environments such as the DeepMind Control Suite Tassa et al. (2018), for example, lack meaningful sub-action interactions Beeson et al. (2024).

## B.2   Varying Dimensionality

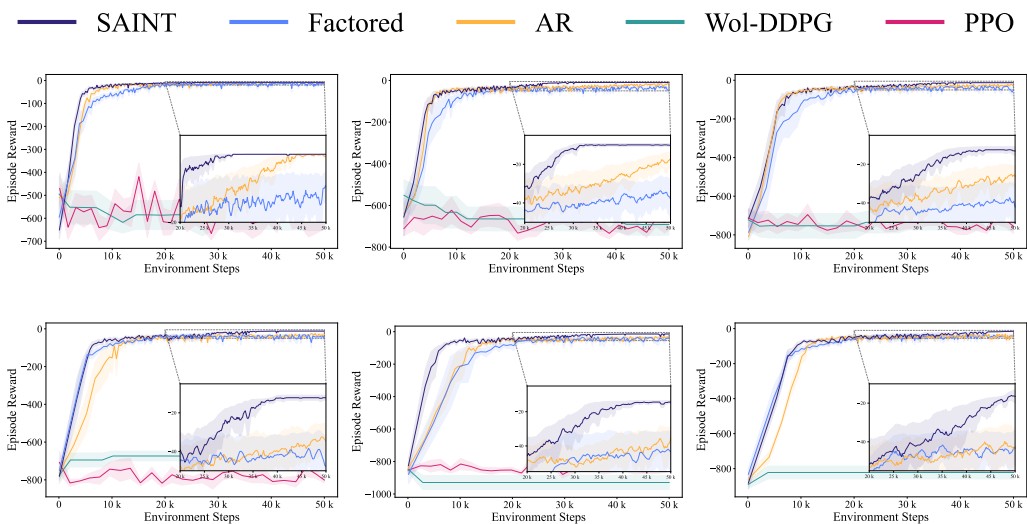

Figure 6: Learning curves in CoNE environments as the number of sub-action dimensions increases from 7 to 12 (corresponding to joint action spaces ranging from $\sim$ 16k to $\sim$ 17M actions). SAINT consistently achieves higher final rewards than all baselines, with its advantage widening in higher-dimensional settings. Factorized and autoregressive baselines struggle to scale beyond moderate dimensions, while Wol-DDPG and A2C fail to learn meaningful policies across all tasks. Results are averaged over 5 seeds; shaded regions indicate one standard deviation.

Figure 6 provides the full learning curves corresponding to the results in Table 1, which reports performance in CoNE as the number of sub-action dimensions increases. As dimensionality grows, the joint action space expands exponentially — from roughly 16 thousand to nearly 17 million possible joint actions.

Across all settings, SAINT consistently outperforms baselines. Notably, SAINT maintains stable learning dynamics and low variance even at the largest scales, whereas factorized and autoregressive baselines generally plateau. Wol-DDPG and A2C fail to learn viable policies in any configuration,

highlighting their inability to handle large, unordered combinatorial action spaces. These results underscore SAINT's scalability and its robustness to increasing combinatorial complexity.

## B.3 VARYING DEPENDENCE

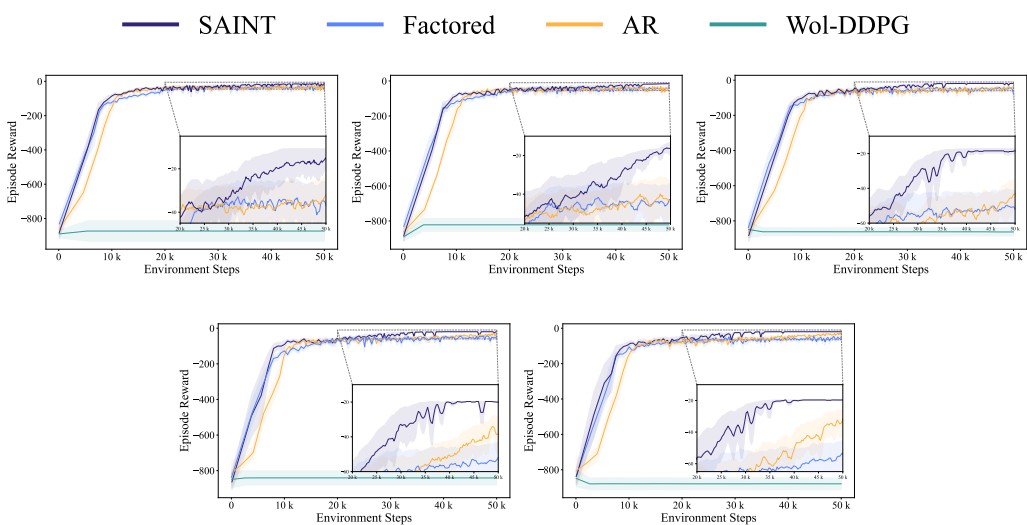

Figure 7: Learning curves in the 12-D CoNE environment as pit density increases from 10% to 100%, inducing progressively stronger sub-action dependencies. SAINT consistently outperforms all baselines, maintaining stable performance even as coordination requirements become increasingly stringent. Factored and autoregressive baselines generally plateau, while Wol-DDPG fails to learn meaningful policies. Results are averaged over 5 seeds; shaded regions denote one standard deviation.

Figure 7 shows the full learning curves corresponding to the results in Table 2, which reports performance in the 12-D CoNE environment as sub-action dependence increases via pit density. As more interior states are occupied by pits, successful navigation requires greater coordination among sub-actions to avoid failure states.

SAINT maintains stable learning and strong final performance across all pit densities, even as coordination requirements grow substantially. Factorized policies degrade, while autoregressive policies consistently underperform relative to SAINT. Wol-DDPG fails to make progress in any environment. A2C was excluded from this experiment due to the computational intractability of modeling such a large discrete action space (nearly 17 million actions) with a flat categorical distribution. These results highlight SAINT's robustness to state-dependent sub-action dependencies.

# C    STATE CONDITIONING

We compare SAINT's pre-attention FiLM-based state conditioning to four alternative mechanisms: (1) applying cross-attention to the state before self-attention, (2) applying cross-attention to the state after self-attention, (3) interleaving cross-attention and self-attention layers, and (4) appending the state as an additional token within the sub-action self-attention block.

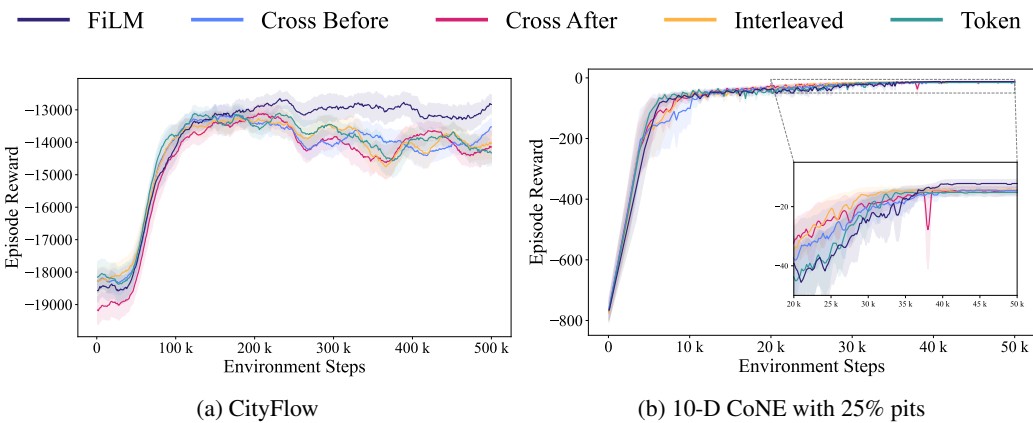

(a) CityFlow              (b) 10-D CoNE with 25% pits

Figure 8: Comparison of state conditioning strategies in CityFlow (left) and 10-D CoNE (right). SAINT's pre-attention FiLM-based conditioning outperforms alternatives more clearly in CityFlow, while all strategies perform similarly in CoNE. Results are averaged over 5 seeds; shaded regions indicate one standard deviation.

As shown in Figure 8, FiLM achieves higher final reward and exhibits more stable learning than the alternatives in CityFlow. In the 10-dimensional CoNE environment, all state conditioning strategies perform comparably, with pre-attention FiLM-based conditioning achieving slightly better final performance. These results suggest that while multiple conditioning mechanisms are viable, FiLM offers an advantage and may contribute to more stable training dynamics.

# D ROBUSTNESS TO OFFLINE RL TRAINING OBJECTIVE

The results in Section 5.4 show that SAINT's set-based, permutation-invariant architecture learns effective representations for offline RL in combinatorial action spaces when trained with an BCQ objective. To assess whether these advantages persist across different offline RL methods, we also evaluate SAINT with two additional objectives, Advantage Weighted Actor Critic (AWAC) (Nair et al., 2020) and Implicit Q-learning (IQL) (Kostrikov et al., 2021). These experiments use the same `medium-expert` datasets considered in Section 5.4 and follow an identical controlled protocol: for each algorithm, we instantiate factorized, autoregressive, and SAINT-based policy parameterizations while keeping the critic, training procedure, and hyperparameters fixed.

## D.1 AWAC

| Task | SAINT | Factored | AR |
|---|---|---|---|
| cheetah | 668.5 ± 21.9 | 657.5 ± 25.9 | 646.6 ± 22.4 |
| finger | 638.6 ± 324.7 | 1.0 ± 1.0 | 1.1 ± 1.4 |
| humanoid | 694.7 ± 29.1 | 639.4 ± 29.2 | 682.9 ± 36.3 |
| quadruped | 837.0 ± 34.9 | 834.2 ± 37.6 | 822.3 ± 46.1 |
| dog | 543.9 ± 60.3 | 423.0 ± 51.7 | 449.5 ± 43.3 |
| **Average** | 676.5 | 511.0 | 520.5 |

Table 5: Mean ± std performance on offline DM Control tasks with AWAC variants.

## D.2 IQL

| Task | SAINT | Factored | AR |
|---|---|---|---|
| cheetah | 627.5 ± 39.6 | 588.7 ± 48.0 | 615.9 ± 37.8 |
| finger | 847.0 ± 13.6 | 841.2 ± 16.2 | 843.5 ± 16.1 |
| humanoid | 613.1 ± 58.9 | 589.9 ± 40.8 | 568.2 ± 55.5 |
| quadruped | 863.7 ± 30.5 | 863.2 ± 36.5 | 857.0 ± 30.8 |
| dog | 596.1 ± 53.2 | 497.8 ± 35.9 | 539.8 ± 33.6 |
| *Average Return* | 709.5 | 676.2 | 684.9 |

Table 6: Mean ± std performance on offline DM Control tasks with IQL variants.

Tables 5 and 6 show that SAINT achieves the strongest performance across all domains under both AWAC and IQL, consistent with the BCQ results in the main text. This indicates that the benefits of modeling sub-action interactions extend across offline RL objectives and are not tied to a specific learning algorithm.

# E  SAINT'S COMPUTATION COST

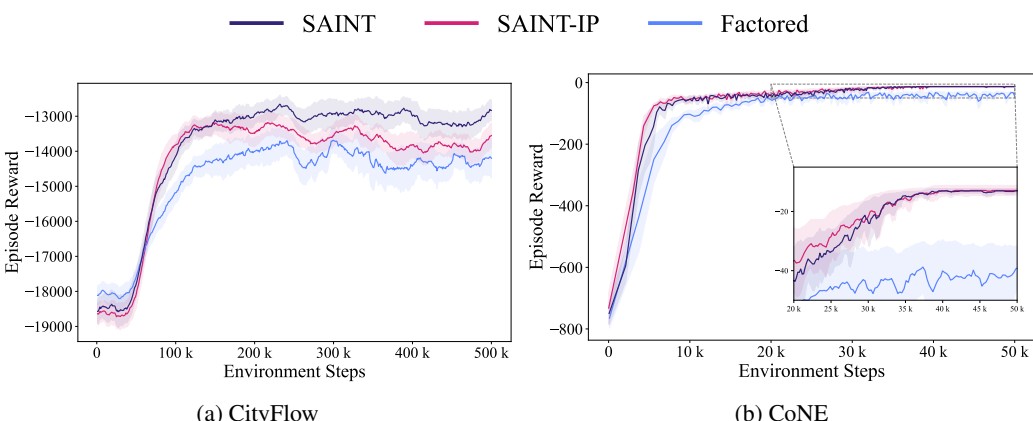

(a) CityFlow                                  (b) CoNE

Figure 9: Learning curves comparing SAINT, SAINT with inducing points (SAINT-IP), and the pure factorized baseline in CityFlow and CoNE. Despite higher per-step computation cost, both SAINT and SAINT-IP reach the factored baseline's final performance much faster and achieve higher final rewards. In CityFlow, SAINT-IP's policy is worse than SAINT's, but still outperforms the baseline. Results are averaged over 5 seeds; shaded regions denote one standard deviation.

Figure 9 shows training curves for SAINT, SAINT with inducing points (SAINT-IP), and the factorized baseline in the CityFlow and CoNE environments. While SAINT and SAINT-IP incur higher per-step computational costs due to the Transformer blocks, both methods achieve the factorized baseline's final performance in less total training time. This reflects their ability to reach performant policies with fewer training episodes. In CityFlow, SAINT-IP exhibits a degradation in asymptotic reward relative to SAINT, but still outperforms the factorized baseline. These results illustrate that the overhead of modeling sub-action dependencies can be offset by more efficient use of training experience.

## F  ROBUSTNESS TO ARCHITECTURAL HYPERPARAMETERS

We systematically evaluated SAINT's sensitivity to architectural hyperparameters by sweeping over the number of self-attention blocks $\{1, 3, 5\}$ and attention heads $\{1, 2, 4, 8\}$, yielding twelve configurations.

| Configuration | Mean Return |
|---|---|
| 1 block × 1 head | $-13376.6 \pm 973.3$ |
| 1 block × 2 heads | $-13416.6 \pm 1003.6$ |
| 1 block × 4 heads | $-13058.6 \pm 725.5$ |
| 1 block × 8 heads | $-13744.5 \pm 900.6$ |
| 3 blocks × 1 head | $\mathbf{-12834.6 \pm 499.8}$ |
| 3 blocks × 2 heads | $-13195.0 \pm 618.8$ |
| 3 blocks × 4 heads | $-13209.5 \pm 778.8$ |
| 3 blocks × 8 heads | $-13156.5 \pm 581.9$ |
| 5 blocks × 1 head | $-13320.7 \pm 831.7$ |
| 5 blocks × 2 heads | $-13658.8 \pm 855.6$ |
| 5 blocks × 4 heads | $-13664.3 \pm 961.5$ |
| 5 blocks × 8 heads | $-13664.8 \pm 822.3$ |
| Factored PPO | $-14200.5 \pm 1127.4$ |
| AR PPO | $-13995.9 \pm 789.4$ |
| Standard PPO | $-14442.5 \pm 1180.0$ |

Table 7: Mean episodic return $\pm$ standard error on CityFlow Irregular. We varied the number of attention blocks $\{1, 3, 5\}$ and the number of attention heads $\{1, 2, 4, 8\}$, for a total of 12 configurations. All SAINT variants outperform Factored, AR, and PPO baselines. The best SAINT configuration is in **bold**, the worst is in *italics*.

A consistent pattern emerges in Table 7. Moderate depth (3 blocks) with 2–4 heads yields strong and stable performance, while very high head counts (8) tend to degrade results. Crucially, every SAINT variant outperforms all baselines, including Factored PPO, AR-PPO, and standard PPO. This robustness implies that SAINT's architectural advantages are not narrowly tied to a specific hyperparameter regime but instead generalize across a broad design space. Careful tuning can yield an additional 5–10% improvement, yet even suboptimal choices consistently achieve better outcomes than state of the art methods.

