# OpenReview forum: "SAINT: Attention-Based Policies for Discrete Combinatorial Action Spaces"
_ICLR.cc/2026/Conference — Submitted to ICLR 2026_

### Official Review · Reviewer_vT94 · 2025-10-28

**Soundness:** 2
**Presentation:** 3
**Contribution:** 2
**Rating:** 2
**Confidence:** 4

**Summary:**

The paper proposes the Sub-Action Interaction Network (SAINT), a novel policy architecture for discrete combinatorial action spaces. The authors identify that existing methods, such as factorized and autoregressive policies, are limited. Factorized methods fail to model sub-action dependencies, while autoregressive methods impose an arbitrary, permutation-variant order. SAINT addresses this by treating the $A$ sub-actions as an unordered set. The architecture uses a learnable embedding for each sub-action index, conditions it on the global state $s$ via FiLM, and then processes the resulting set of vectors through a Transformer (self-attention) that omits positional encodings to achieve permutation-equivariance. The final context-aware vectors are decoded in parallel to produce per-sub-action distributions, and the entire policy is trained with a standard algorithm (PPO). The method is evaluated on CityFlow, a synthetic navigation environment (CoNE), and three discretized MuJoCo tasks.

**Strengths:**

1.  **Clarity:** The paper is written with outstanding clarity, making the problem, prior work, and the proposed method very easy to understand.
2.  **Problem Formulation:** The authors correctly identify a key limitation of existing approaches, namely the rigid and often incorrect inductive bias of a fixed autoregressive ordering.
3.  **Architectural Fit:** The idea of using a permutation-equivariant architecture is an elegant and principled solution for the *specific class of problems* where sub-actions are, in fact, an unordered set (e.g., selecting a set of traffic signals to turn green).

**Weaknesses:**

1.  **Incremental Novelty:** The technical contribution is thin. The method consists of a known neural architecture (self-attention on an unordered set) plugged into a standard, on-policy algorithm (PPO). This is an exercise in architectural engineering, not a new method, and its novelty is limited.
2.  **Fundamentally Questionable Inductive Bias:** The paper's entire motivation rests on the assumption that permutation-equivariance is a *universally desirable* property. This is a strong and, in many cases, incorrect inductive bias. For many (if not most) complex control tasks, sub-actions are *not* interchangeable. For example, in a Humanoid, the action dimensions for the left leg and right leg have distinct, non-exchangeable identities. In such tasks, SAINT's bias is as incorrect as a poorly chosen autoregressive order. The paper fails to discuss this critical limitation, instead presenting the bias as an unequivocal good.
3.  **Weak and Narrowly-Scoped Empirical Validation:** The experiments are not convincing.
    * The primary benchmarks, CityFlow and CoNE, are not standard, challenging testbeds for complex control. They appear to be simple or synthetic environments selected specifically to align with the method's permutation-invariant assumption.
    * The MuJoCo evaluation, which is the most critical, is cherry-picked (only 3 envs) and ultimately unconvincing. In Figure 3a (HalfCheetah), SAINT's performance is **indistinguishable** from the simpler factorized and autoregressive baselines. This result, from the authors' own experiments, directly contradicts the paper's claims and suggests the significant architectural and computational overhead of SAINT is often unwarranted.
    * The **omission** of harder, standard, high-dimensional tasks (e.g., discretized Humanoid) is a major flaw. These tasks have strong, state-dependent, and, crucially, **non-equivariant** action dependencies. Demonstrating performance on such tasks is essential for a paper claiming a general solution for combinatorial action spaces. And many prevailing VLA tasks / datasets are perfect test bed for this paper, I encourage authors test their algo on Open X-Embodiment like dataset. I do notice the in need of simulation env, their are already many of Isaac Sim / Unity based industrial level envs and leave the research for authors.

**Questions:**

1.  The premise of the paper is that permutation-equivariance is superior to a fixed autoregressive order. How do you defend this assumption for the large class of problems (e.g., Humanoid) where sub-actions are *not* interchangeable and have fixed, distinct identities?
2.  Why were more complex, high-dimensional MuJoCo tasks like Humanoid omitted from the evaluation? These are standard benchmarks and would provide a far more convincing test of the method's capabilities and limitations. (It's common practice to discrete continuous action space)
3.  In Figure 3a (HalfCheetah), SAINT provides no measurable benefit over the baselines. How do you reconcile this with the paper's central claim? Does this not suggest that the method's added complexity is often unnecessary?

---

> ### Author Response · Authors · 2025-11-20
>
> We thank the reviewer for the careful and detailed assessment. We appreciate your positive comments on the clarity of the paper, the problem formulation, and the suitability of a permutation-equivariant policy for unordered sub-action sets. Your primary concern is the inductive bias of SAINT’s permutation-equivariant architecture. This reflects an issue of exposition and clarity rather than intent.We agree that permutation-equivariant policies are *not* universally preferable; they are designed for domains where sub-action indices are arbitrary and the important structure comes from interactions among sub-actions. In the revised manuscript, we make this scope explicit and provide additional empirical evidence supporting SAINT’s efficacy in its intended domains. We hope these clarifications are reflected in your final evaluation.
>
> ### W1: Novelty and contribution
>
> Although SAINT uses standard building blocks such as self-attention and common RL objectives, our contribution lies in the **identification and principled solution of a fundamental representation learning problem** in RL.
>
> As discussed in Section 2, several influential works (e.g., Decision Transformer) have similarly advanced the field not by introducing new primitives, but by **recasting a core RL problem under a new modeling paradigm** and applying existing architectures in a novel and meaningful way. Our work contributes in this same manner.
>
> Two central ideas underlie our contribution:
>
> 1. **Core formulation of the problem.** We frame learning in discrete combinatorial action spaces as a **representation learning problem over unordered sets of sub-actions**, moving beyond the standard dichotomy between fully factorized and fixed-order autoregressive policies.
>
> 2. **Principled architectural realization.** SAINT operationalizes this formulation through permutation-equivariant self-attention over sub-action embeddings, enabling the model to learn the internal structure of the action space in both online and offline (see `W3.3/Q2`) RL settings.
>
> The strength of this formulation is supported by our empirical results, including **new experiments** on complex, non-symmetric tasks such as Humanoid and Dog (see `W3.3/Q2`), where SAINT consistently outperforms factorized and autoregressive baselines. These findings demonstrate that SAINT provides not a minor architectural adjustment, but a practical and compelling representational paradigm for combinatorial control.
>
>
> ### W2/Q1: Inductive bias and scope of permutation-equivariance
>
> We appreciate the reviewer's careful discussion of inductive bias. Our intent is **not** to claim that permutation-equivariant policies are universally preferable, but to propose them as a natural model for a broad class of combinatorial control problems where the indexing of sub-actions is arbitrary or only weakly structured.
>
> Importantly, SAINT's permutation-equivariance applies to **indices, not semantics**. The architecture ensures that if sub-actions are relabeled (for example, by permuting the order of traffic signals), the policy’s behavior transforms consistently under the same permutation. This does **not** imply that sub-actions are interchangeable or share identical roles. Distinct roles — such as different intersections or joints — are captured through the learned embeddings and state conditioning, allowing SAINT to assign different behaviors to different sub-actions.
>
> For this reason, while we do not view permutation-equivariance as universally desirable, we do not consider it inherently inappropriate when action dimensions have fixed semantics. To clarify our claim, **SAINT is intended for settings in which the ordering of sub-actions is arbitrary or weakly meaningful, and the primary challenge is to model their interactions.** To make this scope explicit, we updated Section 1 (line 47):
>
> > Our work targets precisely these settings: combinatorial action spaces wherein sub-action indexing is arbitrary or only weakly meaningful, and the fundamental structure lies in sub-action interactions rather than in any prescribed ordering.
>
> And Section 6 (line 524):
>
> > Next, SAINT is designed for domains wherein a joint action is naturally represented as a set of parallel sub-actions for which indexing is arbitrary or weakly structured. In settings with highly structured and known priors, a fully permutation-equivariant prior may not be the most effective representation. In such settings hybrid architectures that combine structured embeddings with partial equivariance are a promising direction for future work.
>
> Accordingly, we do **not** present permutation-equivariance as an unconditional advantage for all control problems. Instead, SAINT offers a principled inductive bias for a well-defined and practically important subset of combinatorial action spaces.

---

> ### Author Response · Authors · 2025-11-20
>
> ### W3.1: Choice of benchmark environments
>
> We respectfully disagree with the characterization of our benchmarks as simple or insufficiently challenging. Our experiments are specifically designed to evaluate architectures under genuine combinatorial pressure, which requires environments where we can directly control the sources of combinatorial difficulty — action dimensionality, interaction strength, and context dependence — without confounding from unrelated factors.
>
> CityFlow is not a toy environment. It is a widely used traffic simulator with realistic network dynamics. Each joint action specifies coordinated signal phases across multiple intersections, yielding a challenging control problem in a congested network. Importantly, equivariance is an inherent property of the domain rather than something we impose. Reindexing intersections leaves the physical control problem unchanged, so any policy that produces different outputs under such a relabeling is mis-specified. All baselines operate under the same environment and objective, so SAINT’s improvements reflect true representational differences rather than artifacts of environmental design.
>
> CoNE is synthetic, but far from simple. It was created specifically to study combinatorial action spaces and to reveal the strengths and weaknesses of competing architectural assumptions. CoNE served as the sole benchmark in BraVE [1], a recent NeurIPS paper, underscoring its legitimacy and difficulty as an evaluation domain. In our largest configuration, CoNE contains more than **200 million states** and nearly **17 million joint actions** per state, with state-dependent pit regions that create strong, context-specific sub-action interactions. This level of control allows systematic stress-testing as dimensionality and dependency strength increase.
>
> Finally, we note that our new results on complex, non-symmetric tasks such as `Humanoid` and `Dog` (see W3.3/Q2) show that SAINT's utility extends beyond these environments.
>
>
> ### W3.2/Q3: Performance in environments with weak sub-action dependencies
>
> As noted in `W2/Q1`, we do not claim that SAINT yields improvements in every possible setting. The discretized MuJoCo tasks were included deliberately as a **weak-dependence regime** (Section 5.3, line 393), where prior work [2] and our own analysis indicate that sub-action interactions are far less pronounced than in CityFlow or CoNE. These tasks serve to evaluate performance when SAINT's set-based, permutation-equivariant inductive bias is less critical.
>
> We have made this point more explicit in Section 5.3 (line 399):
>
> > This setting thus provides a useful test of SAINT’s generality, elucidating whether the architectural overhead of self-attention remains beneficial when sub-action dependencies are weak or whether simpler factorized policies suffice.
>
> Under this interpretation, the HalfCheetah result in Figure 3a is not unexpected: SAINT, the factorized baseline, and the autoregressive baseline obtain nearly identical returns. This does not contradict our claims; it reflects that in settings with limited interaction structure, all three architectures are comparably expressive. Hopper and Walker2D are more informative — SAINT achieves **clear improvements in final return** under the same training protocol, showing that even moderate interaction structure can benefit from a set-based representation.
>
> Regarding the number of MuJoCo tasks, we selected three environments that share a consistent discretization protocol while differing in morphology. This choice reflects computational constraints rather than selective reporting. In direct response to the reviewer’s request for higher-dimensional and non-symmetric domains, we added **new offline evaluations on discretized Humanoid and Dog** (see W3.3/Q2), where SAINT again outperforms factorized and autoregressive policies.
>
> Finally, the claim that SAINT's architectural overhead is often unwarranted is not supported by the broader results. Across all online and offline settings — spanning strong, moderate, and weak sub-action dependencies — SAINT matches or exceeds the baselines. A single environment in which all methods perform similarly (HalfCheetah) does not undermine the consistent pattern observed across the remaining tasks.

---

> ### Author Response · Authors · 2025-11-20
>
> ### W3.3: Omission of high-dimensional MuJoCo tasks
>
> As described in `W3.2/Q3`, the MuJoCo control suite does not naturally exhibit the strong sub-action interactions that SAINT is designed to model. We agree, however, that high-dimensional MuJoCo tasks are valuable for evaluating SAINT’s generality. Based on the reviewer's suggestion, we conducted **new experiments**. In this evaluation, we apply SAINT within BCQ, AWAC, and IQL on the `medium-expert` datasets introduced by Beeson et al. [1], derived from DM Control tasks in which each continuous action dimension is discretized into a finite set. The resulting sub-action counts range from 6 (`cheetah`) to 39 (`dog`), enabling controlled comparison across both low- and high-dimensional action spaces. Two of these environment (`humanoid` and `dog`) directly match the reviewer’s request for challenging, high-dimensional domains.
>
> We chose an offline RL setting for these experiments because a separate reviewer (yLKc) emphasized the importance of offline learning. The DM Control testbed therefore allowed us to **address both concerns simultaneously**: (1) evaluating SAINT on harder, higher-dimensional locomotion tasks, and (2) demonstrating its applicability to offline RL within a single, coherent experimental framework.
>
> These additions appear in a new subsection, Section 5.4 (line 410), with full results in a new Appendix section (Section D). For convenience, we report the results below:
>
> **BCQ**
>
> | Task        | SAINT                | Factored         | AR               |
> |-------------|----------------------|------------------|------------------|
> | cheetah     | **676.1**$\pm$30.9   | 629.6$\pm$36.5   | 629.6$\pm$38.8   |
> | finger      | **809.6**$\pm$29.0   | 692.3$\pm$71.7   | 762.1$\pm$50.9   |
> | humanoid    | **676.5**$\pm$48.2   | 594.1$\pm$47.7   | 592.4$\pm$58.5   |
> | quadruped   | **851.5**$\pm$32.5   | 835.2$\pm$52.0   | 692.2$\pm$99.7   |
> | dog         | **586.1**$\pm$30.5   | 415.2$\pm$40.4   | 423.3$\pm$72.6   |
> | **Average** | **720.0**            | 633.3            | 619.9            |
>
> **AWAC**
>
> | Task        | SAINT                  | Factored                | AR                   |
> |-------------|------------------------|-------------------------|----------------------|
> | cheetah     | **668.5**$\pm$21.9     | 657.5$\pm$25.9          | 646.6$\pm$22.4       |
> | finger      | **638.6**$\pm$324.7    | 1.0$\pm$1.0             | 1.1$\pm$1.4          |
> | humanoid    | **694.7**$\pm$29.1     | 639.4$\pm$29.2          | 682.9$\pm$36.3       |
> | quadruped   | **837.0**$\pm$34.9     | **834.2**$\pm$37.6      | 822.3$\pm$46.1       |
> | dog         | **543.9**$\pm$60.3     | 423.0$\pm$51.7          | 449.5$\pm$43.3       |
> | **Average** | **676.5**              | 511.0                   | 520.5                |
>
> **IQL**
>
> | Task              | SAINT                  | Factored                 | AR                      |
> |-------------------|------------------------|--------------------------|-------------------------|
> | cheetah           | **627.5**$\pm$39.6     | 588.7$\pm$48.0           | 615.9$\pm$37.8          |
> | finger            | **847.0**$\pm$13.6     | **841.2**$\pm$16.2       | **843.5**$\pm$16.1      |
> | humanoid          | **613.1**$\pm$58.9     | 589.9$\pm$40.8           | 568.2$\pm$55.5          |
> | quadruped         | **863.7**$\pm$30.5     | **863.2**$\pm$36.5       | **857.0**$\pm$30.8      |
> | dog               | **596.1**$\pm$53.2     | 497.8$\pm$35.9           | 539.8$\pm$33.6          |
> | *Average Return*  | **709.5**              | 676.2                    | 684.9                   |
>
> Across all three offline algorithms, SAINT yields **consistent and substantial improvements** over the factorized and autoregressive baselines. The gains are particularly pronounced in the highest-dimensional domain (`dog`, 39 sub-actions). While the factored and autoregressive variants of IQL perform competitively on some tasks, SAINT attains the highest overall average return. These results provide strong empirical evidence that SAINT’s set-based, permutation-invariant architecture learns an effective representation for offline RL in combinatorial action spaces.
>
> We reiterate that, as clarified in `W2/Q1` and reflected in our revised Introduction and Discussion, our goal is to provide a principled representation learning framework for the broad and practically important class of problems in which a joint action decomposes into coordinated sub-actions and where the interaction structure among those sub-actions is central to effective decision-making. The new experiments — particularly on `humanoid` and `dog` — were added to evaluate SAINT in the kinds of complex, high-dimensional domains highlighted by the reviewer.
>
>
> [1] Landers et al. "BraVE: Offline Reinforcement Learning for Discrete Combinatorial Action Spaces"
>
> [2] Beeson et al., "An Investigation of Offline Reinforcement Learning in Factorisable Action Spaces"

---

> > ### Comment · Reviewer_vT94 · 2025-11-21
> > **Acknowledgement of Rebuttal and Score Update**
> >
> > I thank the authors for their detailed response and the substantial effort put into conducting new experiments on high-dimensional offline RL tasks (e.g., Humanoid, Dog). I acknowledge that these additional results demonstrate the method's efficacy in higher-dimensional settings and clarify the intended scope of the permutation-invariant inductive bias.
> >
> > In light of these improvements and the effort involved, I am raising my score to **4**.
> >
> > However, I maintain my recommendation to borderline reject. My primary concern regarding the nature of the contribution remains unresolved. Fundamentally, applying self-attention to sub-action embeddings to model dependencies is a straightforward architectural application of existing Transformer components to RL. While the authors cite the foundational Set Transformer work (Lee et al., 2019), the application of permutation-invariant attention to handle unordered sets is already a well-established design pattern in closely related fields. For instance, it is standard in Multi-Agent RL for handling unordered agents (PIC, Liu et al., CoRL 2019) and in Neural Combinatorial Optimization (Kool et al., ICLR 2019).
> >
> > Given this context, I view SAINT as a logical engineering extension—applying known "Set Transformer" principles to the policy head—rather than the significant conceptual or algorithmic novelty typically required for ICLR.

---

> > > ### Author Response · Authors · 2025-11-21
> > >
> > > We sincerely thank the reviewer for their continued engagement, for acknowledging the added high-dimensional experiments, and for raising their score. We appreciate the opportunity to clarify the reviewer's remaining concern about the nature of the contribution and its relation to prior work.
> > >
> > > We agree that SAINT is built from standard architectural components. Our claim is not that these primitives are new, but that there is a specific representational problem in RL — policies over discrete combinatorial action spaces — that benefits from being modeled as sets of interacting sub-actions, and that SAINT provides a principled and general solution to this problem.
> > >
> > > Prior work highlighted by the reviewer uses similar architectural primitives as SAINT but in fundamentally different functional roles. In PIC (Liu et al.), permutation-invariant processing appears only in a **critic** network that aggregates inputs to a scalar and does not define how a policy should represent or parameterize a combinatorial action. In neural combinatorial optimization methods such as Kool et al., attention is embedded within a task-specific **autoregressive** decoder that constructs a solution sequentially according to a fixed decoding order, so the combinatorics arise from the **sequential horizon** rather than from a single joint action.
> > >
> > > None of these methods treat the **joint action at one timestep in a Cartesian-product action space** as the structured object to be modeled. SAINT, by contrast, represents $\mathbf{a} = (a_1,\dots,a_A)$ as an unordered set of sub-action slots with state-conditioned embeddings, applies permutation-equivariant self-attention to learn sub-action interactions, and decodes all sub-action distributions in parallel, yielding a joint action distribution without imposing any ordering. This joint-action-as-set formulation — realized as a policy-level, order-free, permutation-equivariant architecture, and validated across traffic control, navigation, and high-dimensional locomotion — is the core novelty of the paper.
> > >
> > > We recognize that SAINT is architectural in the sense that it introduces no new learning rules or Transformer variants. However many impactful papers (including those referenced by the reviewer) apply a conceptually simple and generalizable principle to a new domain with significant effect. We position SAINT in this light. The work goes beyond a small architectural adjustment, **it isolates a practically important representational problem, provides a principled solution, instantiates that solution in a general-purpose architecture, and demonstrates its effectiveness across a wide range of RL settings**. Recent interest in discrete combinatorial action spaces further underscores the need for a clear, well-defined benchmark, and we expect SAINT’s principled and concise formulation to serve as a strong reference point for future work in this area.

---

### Official Review · Reviewer_yLKc · 2025-11-01

**Soundness:** 3
**Presentation:** 3
**Contribution:** 3
**Rating:** 6
**Confidence:** 3

**Summary:**

Problem:

The paper addresses discrete combinatorial action spaces (each joint action contains multiple coordinated subactions) in RL. Taking the Cartesian product of subactions and applying standard RL is intractable as the action space grows exponentially, while existing simplifications (factorization or autoregressive sequencing) fail to reliably capture complex interaction effects between subactions.


Approach:

SAINT (Sub-Action Interaction Network using Transformers) models actions as unordered sets of subactions and uses self-attention (transformers) conditioned on the global state to capture their dependencies. The design is permutation-invariant, scalable, and compatible with common RL policy optimization algorithms (e.g., PPO, A2C).


Overall assessment (I am not too familiar with the relevant literature solving the same problem):

The proposed approach, SAINT, takes an important step to address problems where lots of decisions must be made together and coordinated, especially in big systems. It’s most useful where those decisions really influence each other. It’s less useful when each decision is simple and doesn’t depend on the others. Overall, this makes RL smarter for complicated real-world tasks. The evaluation is thorough and convincing.

**Strengths:**

- The proposed approach can model complex, context-sensitive dependencies in large action spaces. It is permutation invariant, i.e., naturally fits unordered action compositions.

- The evaluation conducted is extensive and compelling. I appreciate the ablations. Experiments show that the proposed approach consistently outperforms baselines on diverse tasks: state-independent (traffic control), state-dependent (navigation), and weakly dependent (discretized MuJoCo). The scalability of the approach is impressive.

**Weaknesses:**

- The approach may be less justified for low-dimensional or weakly structured domains.

Suggestions:

- Since combinatorial action spaces are common in offline RL (e.g., healthcare), systematic analysis in off-policy contexts could further establish SAINT's utility.

**Questions:**

- Did you consider environments where sub-action sets themselves change dynamically (e.g., road closures or reconfiguration in traffic networks)? How extensible is the permutation-invariant approach in such cases?

- Can SAINT be adapted effectively for offline RL, especially in settings with sparse and partial combinatorial action coverage?

---

> ### Author Response · Authors · 2025-11-20
>
> We thank the reviewer for the thoughtful and constructive assessment. We appreciate your positive remarks regarding SAINT's scalability and the strength of the empirical evaluation. Your suggestions regarding dynamic sub-action sets and offline RL provided helpful guidance and directly motivated a **new set of experiments** in the revision. We hope these additions address your concerns and that you will consider raising your score.
>
> ### W1: Applicability to low-dimensional or weakly structured domains
>
> While SAINT's advantages are most pronounced in domains with strong sub-action dependencies, we also evaluated it in settings where these dependencies are weak, using the discretized MuJoCo tasks (Section 5.3). As noted on line 393, these environments have challenging underlying dynamics despite the weak sub-action dependencies. Figure 3 shows that SAINT learns faster and achieves higher returns in Hopper and Walker2D, indicating that the set-based representation remains effective even in this regime.
>
> We have clarified this point in Section 5.3 (line 399):
>
> > This setting thus provides a useful test of SAINT's generality, clarifying whether the architectural overhead of self-attention remains beneficial when sub-action dependencies are weak or whether simpler factorized policies suffice.
>
>
> ### Q1: Applicability to dynamic sub-action sets
>
> This is an important practical question. SAINT does not rely on a fixed action-set size as the Transformer operates on a set of sub-action embeddings, and the policy head produces a distribution only over the sub-actions that are valid at a given timestep.
>
> In practice, one can define embeddings for a superset of all possible sub-action indices and restrict attention and decoding to the subset available at each timestep. For example, in a traffic-control setting, a road closure simply corresponds to marking certain sub-actions as inactive; SAINT omits those tokens (or masks their logits) without requiring any architectural changes.
>
> Because self-attention is permutation-equivariant and acts over sets, removing unavailable sub-actions does not interfere with the model's ability to reason about the remaining sub-actions. This provides a **natural advantage over architectures with a fixed autoregressive order**, which are more brittle to such changes. Removing a sub-action from the middle of an AR model's fixed sequence breaks the chain of conditional dependencies it was trained to rely on, whereas removing an element from SAINT's set is a native operation that does not violate its architectural assumptions.
>
> Thus, while our experiments use a fixed action set, extending SAINT to dynamically changing sub-action sets is a natural direction for future work. We have added the following to Section 6 (line 529):
>
> > Finally, although our experiments assume a fixed set of sub-actions, many real-world domains, such as road closures or reconfigurable network topologies, involve dynamically changing action sets. Because SAINT represents actions as an unordered set, it naturally supports such variability via masking. Systematically evaluating this capability is an important research direction.

---

> ### Author Response · Authors · 2025-11-20
>
> ### Q2: Applicability to offline RL
>
> We agree that offline RL is an important and practical application domain for SAINT. Motivated by your suggestion, we conducted a **new set of experiments** to assess SAINT's effectiveness within standard offline algorithms. Specifically, we evaluate SAINT with BCQ, AWAC, and IQL on the challenging `medium-expert` datasets introduced by Beeson et al. [1], derived from DM Control tasks where each continuous action dimension is discretized into a finite set. The resulting sub-action counts range from 6 (`cheetah`) to 39 (`dog`), enabling controlled comparison across both low- and high-dimensional action spaces.
>
> These additions appear in a new subsection, Section 5.4 (line 410), with full results in a new Appendix section (Section D). For convenience, we report the results below:
>
> **BCQ**
>
> | Task        | SAINT                | Factored         | AR               |
> |-------------|----------------------|------------------|------------------|
> | cheetah     | **676.1**$\pm$30.9   | 629.6$\pm$36.5   | 629.6$\pm$38.8   |
> | finger      | **809.6**$\pm$29.0   | 692.3$\pm$71.7   | 762.1$\pm$50.9   |
> | humanoid    | **676.5**$\pm$48.2   | 594.1$\pm$47.7   | 592.4$\pm$58.5   |
> | quadruped   | **851.5**$\pm$32.5   | 835.2$\pm$52.0   | 692.2$\pm$99.7   |
> | dog         | **586.1**$\pm$30.5   | 415.2$\pm$40.4   | 423.3$\pm$72.6   |
> | **Average** | **720.0**            | 633.3            | 619.9            |
>
> **AWAC**
>
> | Task        | SAINT                  | Factored                | AR                   |
> |-------------|------------------------|-------------------------|----------------------|
> | cheetah     | **668.5**$\pm$21.9     | 657.5$\pm$25.9          | 646.6$\pm$22.4       |
> | finger      | **638.6**$\pm$324.7    | 1.0$\pm$1.0             | 1.1$\pm$1.4          |
> | humanoid    | **694.7**$\pm$29.1     | 639.4$\pm$29.2          | 682.9$\pm$36.3       |
> | quadruped   | **837.0**$\pm$34.9     | **834.2**$\pm$37.6      | 822.3$\pm$46.1       |
> | dog         | **543.9**$\pm$60.3     | 423.0$\pm$51.7          | 449.5$\pm$43.3       |
> | **Average** | **676.5**              | 511.0                   | 520.5                |
>
> **IQL**
>
> | Task              | SAINT                  | Factored                 | AR                      |
> |-------------------|------------------------|--------------------------|-------------------------|
> | cheetah           | **627.5**$\pm$39.6     | 588.7$\pm$48.0           | 615.9$\pm$37.8          |
> | finger            | **847.0**$\pm$13.6     | **841.2**$\pm$16.2       | **843.5**$\pm$16.1      |
> | humanoid          | **613.1**$\pm$58.9     | 589.9$\pm$40.8           | 568.2$\pm$55.5          |
> | quadruped         | **863.7**$\pm$30.5     | **863.2**$\pm$36.5       | **857.0**$\pm$30.8      |
> | dog               | **596.1**$\pm$53.2     | 497.8$\pm$35.9           | 539.8$\pm$33.6          |
> | *Average Return*  | **709.5**              | 676.2                    | 684.9                   |
>
> Across all three offline algorithms, SAINT yields **consistent and substantial improvements** over the factorized and autoregressive baselines. The gains are particularly pronounced in the highest-dimensional domain (`dog`, 39 sub-actions). While the factored and autoregressive variants of IQL perform competitively on some tasks, SAINT attains the highest overall average return. These results provide strong empirical evidence that SAINT's set-based, permutation-invariant architecture learns an effective representation for offline RL in combinatorial action spaces.
>
> We have also updated Section 4.4 (line 225) to explicitly note compatibility with offline RL methods:
>
> > Compatible methods include standard online algorithms such as PPO and A2C, as well as offline approaches such as IQL and AWAC. SAINT also supports selection-based actor updates as in BCQ, where the policy is trained on candidate joint actions drawn from a dataset or proposal distribution.
>
> [1] Beeson et al., "An Investigation of Offline Reinforcement Learning in Factorisable Action Spaces"

---

### Official Review · Reviewer_huDo · 2025-11-06

**Soundness:** 2
**Presentation:** 2
**Contribution:** 2
**Rating:** 4
**Confidence:** 3

**Summary:**

The authors propose a policy architecture that learns representations of combinatorial action space, by treating actions as unordered sets of sub-actions. They show empirically that the proposed method has strong performance.

**Strengths:**

The authors provide ablations showing the robustness of the  proposed method  on varying dimensionality and varying sub-action dependence.

**Weaknesses:**

The proposed method can have high computational costs. when action space is large, the learnable embedding vector e_i has high dimension. Adding state conditioning further increase the dimensionality.

**Questions:**

Why do the authors augment action embedding e_i with the global state, instead of using actions as queries and states as keys? Although the former could capture interactions among sub-actions, it may fail to capture interactions between actions and the state.

Could the authors explain why a categorical distribution is used (first equation in Sec 4.3, instead of the standard softmax?

The baselines in experiments are weak. Could the authors include at least one attention-based policy as a baseline? For existing methods that designed for continuous action space, we  may discretize the continuous actions to accommodate the combinatorial action space.

How many seeds are used in Table 1-2 and in Figure 2?

---

> ### Author Response · Authors · 2025-11-20
>
> Thank you for your careful review. We appreciate your recognition of SAINT's effectiveness and the robustness of our empirical evaluations. The concerns you raised were largely matters of exposition. We have clarified these areas in the revised manuscript and respond to each point in detail below. We are confident the revisions directly resolve the issues you identified, and we hope this is reflected in your final score.
>
> ### W1: Computational cost of SAINT representations
>
> We agree that operating in large combinatorial action spaces introduces additional computation, but the reviewer's concern conflates *action-space size* with *embedding dimensionality*. Specifically, the dimensionality of each sub-action embedding $e_i$ is a fixed hyperparameter $d$. As described in Section 4.1 (line 153), we learn an embedding table $\text{Embed} \in \mathbb{R}^{A \times d}$ and represent each sub-action by $e_i = \text{Embed}(i)$; this dimension $d$ is independent of the cardinality $|A_i|$ or of the size of the joint action space. Increasing the number of sub-actions increases the number of tokens processed by the Transformer, but does not increase the dimensionality of each token.
>
> To make this point more explicit, we have revised Section 4.1 (line 154):
>
> > With $d$ treated as a fixed hyperparameter shared across all sub-actions, each sub-action is represented by a $d$-dimensional embedding independent of its original cardinality $|\mathcal{A}_i|$, yielding a shared space for uniform processing by the subsequent Transformer layers.
>
> Crucially, while SAINT's architecture is more expensive per step than factorization, our experiments show that by capturing complex sub-action dependencies that simpler models cannot, **SAINT learns significantly more efficiently**. As shown in Table 4, SAINT reaches the asymptotic performance of the factorized baseline in less than one-third of the wall-clock time in CityFlow and over 30% faster in CoNE. This indicates that the added computation is outweighed by faster convergence, leading to a net reduction in the time required to find a strong policy.
>
> ### W2/Q1: Computational cost of SAINT state conditioning
>
> This is an important clarification and, in fact, a strength of our approach: FiLM-based state conditioning **does not increase the embedding width**. At each timestep, FiLM processes the global state $s$ once to produce $(\gamma, \beta) = g(s)$ and applies feature-wise modulation $ \tilde e_i = \gamma \odot e_i + \beta$, leaving the dimensionality at $d$. This provides state-aware representations without expanding the vector size or adding additional attention over state.
>
> To clarify this point, we have updated Section 4.1 (line 161):
>
> > Notably, FiLM preserves the $d$-dimensional width of each sub-action embedding, introducing no additional projection dimensions.
>
> Importantly, we **already explicitly evaluated cross-attention–based state conditioning mechanisms** — which *would* introduce additional attention projections and therefore higher computational cost — precisely to test whether their additional flexibility justified the overhead. These variants were consistently slower and less stable than FiLM (Appendix C).
>
> ### Q2: Categorical distribution vs. softmax
>
> Each sub-action distribution is indeed parameterized by softmax-normalized logits. We agree that the shorthand notation in the original text may obscure this. To make the role of the softmax explicit, we have revised Section 4.3 (line 206):
>
> > In the final stage, each context-aware sub-action representation $\mathbf{x}_i$ is passed through a sub-action-specific decision MLP, $f_i: \mathbb{R}^d \rightarrow \mathbb{R}^{K_i}$, which outputs a vector of $K_i$ logits. These logits are then transformed into a probability distribution over the $K_i$ discrete choices for sub-action $i$ via the softmax function. The resulting policy for sub-action $i$ is thus given by:
>
> $$
> \pi\_i(\cdot \mid \mathbf{s}) = \mathrm{Categorical}(\mathrm{softmax}(f\_i(\mathbf{x}_i))).
> $$

---

> ### Author Response · Authors · 2025-11-20
>
> ### Q3: Including an attention-based baseline
>
> We selected baselines that capture the canonical assumptions in combinatorial action spaces: factorized independence, autoregressive ordering, and monolithic flat policies that become intractable at high dimensionality. These baselines are the established standards in this domain, and SAINT consistently and substantially outperforms them.
>
> To our knowledge, SAINT is the first architecture to use attention to model the internal structure of a discrete combinatorial action space. As described in Section 2 (Related Work), existing Transformer-based architectures for control fall into two main categories, neither of which is an appropriate comparator for SAINT:
>
> 1. **Offline trajectory learning.** The dominant use of Transformers in decision making (e.g., Decision Transformer, RT-1) is to model entire $(s, a, r, \dots)$ **trajectories** in an offline setting. These methods are designed to learn *temporal* dependencies from fixed datasets and are therefore not appropriate as baselines for our online experiments.
>
> 2. **Sequential action decompositions.** Some architectures apply Transformers within a timestep but impose a fixed autoregressive ordering over sub-actions. Our autoregressive (AR) baseline is designed to isolate this assumption, and its consistent underperformance across all experiments (Section 5) provides strong evidence that enforcing an arbitrary order is detrimental when the true sub-action dependencies are unordered.
>
> Given this landscape, our baselines capture the dominant structural assumptions in the literature.
>
> ### Q4: Adapting continuous-action methods via discretization
>
> The suggestion to adapt continuous-action methods to a discretized combinatorial space is natural, but doing so requires **committing to one of several structural assumptions** about the action space. The consistent failure of all four canonical assumptions in our experiments motivates SAINT.
>
> The four dominant approaches are:
>
> 1. **Monolithic assumption (flat PPO/A2C).** The flat baseline ignores all compositional structure and treats each joint action as an atomic symbol, leading to methods that are computationally intractable as dimensionality grows.
>
> 2. **Independence assumption (factorized baseline).** This common simplification treats each sub-action as independent by discretizing each dimension separately, preventing the policy from modeling necessary coordination.
>
> 3. **Ordered assumption (autoregressive baseline).** This approach captures dependencies but imposes a fixed, arbitrary ordering over sub-actions. It performs poorly when this imposed order is misaligned with the true dependency structure.
>
> 4. **Action-embedding paradigm (Wol-DDPG).** This approach embeds each joint action as a single continuous vector, preventing the policy from capturing the internal structure of multi-dimensional actions.
>
> The collective failure of these assumptions provides the empirical basis for our work. Each baseline fails because it imposes a rigid — and, in combinatorial action spaces wherein sub-action indexing is arbitrary or only weakly meaningful, ultimately incorrect — prior over the action space. SAINT, by contrast, learns the state-conditioned, permutation-invariant dependency structure directly from data, without committing to independence, a fixed order, or a flat action embedding.
>
> To clarify this point, we have added the following to Section 5 (line 254):
>
> > These four baselines instantiate the dominant structural assumptions used to scale RL to large combinatorial action spaces. The flat policy corresponds to a monolithic model that ignores compositional structure and treats each joint action as an atomic symbol. The factorized policy enforces independent per-dimension decisions, preventing it from modeling necessary coordination between sub-actions. The autoregressive model imposes a fixed sequential ordering over sub-actions, which can be misaligned with the true, permutation-invariant dependency structure. Wol-DDPG embeds each joint action as a single continuous vector, collapsing the internal structure needed to capture interactions among sub-actions. Our experiments in Sections 5.1–5.4 are designed to test whether these structural assumptions remain sufficient when sub-action indexing is arbitrary or only weakly meaningful, or whether a set-based alternative such as SAINT is required.
>
> ### Q5: Number of seeds
>
> Results are averaged over five random seeds, this is specified on line 252.

---

### Author Response · Authors · 2025-11-20

### General response

We sincerely thank all reviewers for their thoughtful and constructive feedback. We are encouraged that they recognized the merits of our work. Reviewers highlighted the strength of our method, noting its "**strong performance**" [huDo], "**robustness**" [huDo], and that SAINT "**takes an important step**" toward principled reasoning in combinatorial action spaces [yLKc]. They further emphasized that our evaluation is "**thorough and convincing**" [yLKc], "**extensive and compelling**" [yLKc], and that our approach "**consistently outperforms baselines on diverse tasks**" [yLKc]. We also appreciate the remarks on the presentation quality, noting "**outstanding clarity, making the problem, prior work, and the proposed method very easy to understand**" [vT94]. Finally, reviewers noted that we "**correctly identify a key limitation of existing approaches**" [vT94] and that our use of a permutation-equivariant architecture is an "**elegant and principled**" way to model sub-action interactions [vT94].

Reviewers raised several important points, which we address comprehensively in our point-by-point responses. In this general remark, we summarize the new experiments and clarifications included in the revised manuscript.

Based on the reviewers' feedback, we conducted a **new set of experiments** and made **additional clarifications** to:

* **Precisely delineate the scope of SAINT's inductive bias**, clarifying when permutation-equivariant structure is appropriate.
* **Evaluate SAINT in high-dimensional, non-symmetric control domains**, adding new results on discretized variants of the `humanoid` and `dog` environments.
* **Demonstrate SAINT's applicability in offline RL**, by integrating SAINT into standard offline algorithms on the discretized DM Control `medium-expert` benchmark.

These additions further strengthen our confidence in SAINT's effectiveness.

We revised the manuscript to address the main themes raised by the reviewers:

* We clarify that SAINT's core contribution is a principled framework for learning representations over unordered sets of interacting sub-actions, and explain how permutation-equivariance over indices —rather than semantics — applies when sub-action indexing is arbitrary or only weakly meaningful [huDo, yLKc, vT94].
* We provide substantial new results demonstrating SAINT's generality and robustness: our offline RL experiments on complex, high-dimensional control tasks directly address requests for broader empirical validation and show that SAINT is effective across a broad range of settings. [yLKc, vT94].
* We clarify the computational characteristics of SAINT's architecture, emphasizing that embedding width is independent of action-space size, FiLM does not increase dimensionality or add attention projections, and SAINT naturally supports dynamic sub-action sets and standard offline RL methods [huDo, yLKc].

### Notes to reviewers

Please note that we use W# to address weaknesses and Q# to address questions in our point-by-point responses. All line numbers refer to the updated manuscript submitted with the rebuttal.

We again thank the reviewers for their thoughtful and constructive feedback. Your comments helped sharpen the presentation and strengthen the contributions of the paper. We would be grateful for the opportunity to address any remaining concerns.

---

### Author Response · Authors · 2025-12-01
**Concise Summary of Rebuttal and Revisions**

Dear Area Chair and Reviewers,

We sincerely thank the Area Chair for their service and the reviewers for their constructive and thoughtful feedback. A concise summary of our rebuttal and revisions is provided below.

### **Concerns & Suggestions (Method & Scope)**

| **Concern** | **Reviewer(s)** | **Our Action** |
| :--- | :--- | :--- |
| **Inductive Bias:** Is permutation-equivariance appropriate in general? | vT94 (W2) | **Clarification:** Emphasized that SAINT is intended for combinatorial action spaces where sub-action indexing is arbitrary or only weakly meaningful, and clarified that permutation-equivariance is not claimed to be universally preferable (Sec. 1, Sec. 6). |
| **Novelty:** Is this simply "Set Transformer + PPO"? | vT94 (W1) | **Clarification:** Made explicit that SAINT introduces the first **action-level, permutation-equivariant policy architecture**, directly modeling a joint action as an interacting set of sub-actions rather than imposing factorization or arbitrary ordering (Sec. 2). |
| **Computational Cost:** Is SAINT too expensive for large action spaces? | huDo (W1) | **Analysis:** Clarified that embedding width is independent of action-space size and showed that SAINT reaches strong performance **$\approx30$% faster in wall-clock time** than factorized baselines due to improved sample efficiency (Sec. 4.1, Sec. 5.5). |
| **Dynamic Action Sets:** Can SAINT handle changing sub-action availability? | yLKc (Q1) | **Clarification:** Explained that SAINT naturally supports dynamic sub-action sets via masking, whereas autoregressive models rely on fixed ordering and are brittle under topology changes (Sec. 6). |

### **Concerns & Suggestions (Evaluation & Empirics)**

| **Concern** | **Reviewer(s)** | **Our Action** |
| :--- | :---| :--- |
| **High-Dimensional, Non-Symmetric Tasks:** Can SAINT handle more complex domains like Humanoid/Dog? | vT94 (W3, Q2) | **New Experiments:** Added results on discretized **Humanoid** and **Dog** (up to 39 sub-actions), where SAINT consistently outperforms factored and autoregressive baselines (Sec. 5.4, App. D). |
| **Offline RL:** Is SAINT effective in off-policy settings? | yLKc (Q2) | **New Experiments:** Integrated SAINT into BCQ, AWAC, and IQL on the DM Control `medium-expert` benchmark; SAINT achieves the **highest average return** across algorithms and tasks (Sec. 5.4, App. D). |
| **Benchmark Choice:** Are CityFlow and CoNE too simple or tailored to SAINT? | vT94 (W3) | **Clarification:** Explained that CityFlow is a realistic traffic-control simulator with inherent index-equivariance, and that CoNE (NeurIPS' 25) enables controlled variation of combinatorial difficulty and complements the new MuJoCo experiments (Sec. 5). |
| **Baselines:** Why no attention-based policy comparisons? | huDo (Q3) | **Clarification:** Noted that existing Transformer-based RL methods operate on **trajectories** or **temporal sequences**, not on the internal structure of a joint action; SAINT is the first attention-based policy designed specifically for **discrete combinatorial action spaces** (Sec. 2). |

We have incorporated these new experiments, analyses, and clarifications into the revised manuscript (highlighted in blue). We are confident that the updates address the reviewers' concerns and further demonstrate that SAINT is a robust and principled framework for combinatorial control.

---

### Meta-Review · Area_Chair_R1EF · 2026-01-13

**Summary:**

This paper proposes SAINT, a policy architecture that represents multi-component actions as unordered sets and models their dependencies via self-attention conditioned on the global state. The empirical study show that SAINT can outperform factorized and autoregressive baselines in settings with stronger sub-action interactions. However, the core contribution remains largely an application of well-established set-attention design patterns, i.e., Set Transformers / permutation-invariant attention, to the policy head, without a clear algorithmic or conceptual advance beyond prior uses of attention for unordered entities in adjacent areas. Moreover,  the performance gains are small or indistinguishable in weak-dependence regimes, and the method introduces potentially significant computational overhead that is not well justified. Overall, I would suggest a borderline reject.

**Reviewer Concerns:**

The rebuttal have clarified the concerns on Humanoid/Dog offline results, wrong inductive bias, and missing hard tasks. However, the technical novelty is still a concern. Specifically, the core idea still looks like a straightforward application of known set-attention ideas to the policy head. In addition, stronger baselines and some modeling-choice clarifications appear unresolved.

**Reviewer Scores:**

Reivewers do not seem like will update their scores.

---

### Decision · Program_Chairs · 2026-01-26

Reject